# Neurons embedded in loop-like motifs act as central hubs for brain-wide integration

Fabrizio Londei[1,2] (iD), Giulia Arena[2,3,4], Lorenzo Ferrucci[3,5] (iD), Francesco Siano[2,3], Encarni Marcos[1] (iD), Francesco Ceccarelli[3] (iD) and Genovesio Aldo[3,4,6] (iD)

[1] *Instituto de Neurociencias, Consejo Superior de Investigaciones Científicas and Universidad Miguel Hernández, Sant Joan d'Alacant, Alicante, Spain*
[2] *Behavioral Neuroscience PhD Program, Sapienza University, Rome, Italy*
[3] *Department of Physiology and Pharmacology, Sapienza University of Rome, Rome, Italy*
[4] *Institute of Biochemistry and Cell Biology (IBBC), National Research Council of Italy (CNR), Monterotondo Scalo, Roma, Italy*
[5] *Department of Wellbeing, Health and Environmental Sustainability, Sapienza University of Rome, Rieti, Italy*
[6] *Department of Pharmaceutical Sciences, University of Piemonte Orientale, Novara, Italy*

Handling Editors: Richard Carson & Bettina Schwab

The peer review history is available in the Supporting Information section of this article (https://doi.org/10.1113/JP289827#support-information-section).

**Abstract figure legend** Neurons form loop-like triplet motifs in which activity originates in *Area_i*, propagates to an external *Area_j* and returns to *Area_i*. Neurons participating in these loops act as major external hubs, forming pairwise assemblies with neurons across multiple other areas (A–N). This identifies loop-like motifs as key integrative elements in large-scale cortical communication, unlike other motifs that lack hub-like properties.

**Fabrizio Londei** earned a master's degree in applied mathematics from Sapienza University of Rome (Italy), where he also completed his PhD in behavioural neuroscience. His research focuses on data analysis and computational neuroscience, particularly investigating the role of cell assemblies at multiple levels, from coding to functional connectivity, and their relationship with dynamic and static coding schemes of information in memory. He is currently a postdoctoral researcher at the Instituto de Neurociencias CSIC-UMH (Spain). **Giulia Arena** is a PhD candidate at Sapienza University of Rome (Italy), where she also earned an MSc in neurobiology, under the supervision of Prof. Aldo Genovesio. In collaboration with IBBC-CNR in Monterotondo (Italy), her research investigates context-dependent population dynamics and large-scale brain network reconfiguration underlying behavioural flexibility and adaptation to specific cognitive demands.

F. Londei and G. Arena have contributed equally to this work.

**Abstract** Efficient cooperation between brain areas requires a dynamic balance between the segregation of region-specific functional roles and information broadcasting. Imaging studies of brain-wide co-ordination cannot reach the single-cell assembly level analysis. In this study, we explored co-ordinative relationships between single neurons across 71 mouse brain regions using the concept of cell assembly as an investigative tool. Cell assemblies can be considered the fundamental brain processing units, and identifying their aggregate structure can provide a high-resolution view of inter-regional co-ordinative relationships. We first examined pairwise co-ordination between areas, and then we investigated higher-order forms of inter-regional connectivity, searching for triplets of neurons. We mainly focused on one functionally relevant motif: the loop-like triplet, modelling a reentrant flow of information, which we hypothesised could represent a core mechanism for the integration of information. We found that this reentrant mode of communication was often asymmetrical between areas and largely unrelated to neuron pair co-ordination. We found that hub neurons, which have a higher-than-average number of co-ordinative relationships with external regions, are consistently and significantly embedded in loop-like assemblies. These findings suggest that this peculiar motif represents a core architectural feature supporting brain-wide integration.

(Received 30 July 2025; accepted after revision 27 January 2026; first published online 5 March 2026)

**Corresponding authors** F. Ceccarelli: Department of Physiology and Pharmacology, Sapienza University of Rome, Piazzale Aldo Moro 5, 00185 Rome, Italy.　Email: francesco.ceccarelli@uniroma1.it

A. Genovesio: Department of Pharmaceutical Sciences, University of Piemonte Orientale, Largo Donegani 2/3 – 28100 Novara.　Email: aldo.genovesio@uniupo.it

**Key points**

- Cell assembly detection allows the identification of motifs of inter-regional co-ordination.
- Loop-like motifs of co-ordination are heterogeneously distributed across the brain.
- External hub neurons are consistently and brain-wide embedded in loop-like motifs.
- Loop-like motifs appear as an integration-oriented structure of co-ordination.

## Introduction

Functional connectivity is defined as the statistical dependence between neurophysiological events happening in different brain regions over time (Friston, 1994) and allows to characterise the relationships between them. Functional connectivity analysis is built on data sampled with several methods, classically electro-encephalography (EEG), magnetoencephalography (MEG), positron emission tomography (PET) and functional magnetic resonance imaging (fMRI), whose acquisitions can be examined through the framework of multiple statistical and mathematical models (for reviews on this refer to Chiarion et al. (2023) and Shahhosseini & Miranda (2022)). However, recent progress in the field of electrophysiology have opened new perspectives. The introduction of Neuropixel probes (Jun et al., 2017), high-density recording electrodes, has enabled the sampling of the activity of up to hundreds of neurons simultaneously (Steinmetz et al., 2019), making functional connectivity graspable even at a single-cell level of resolution.

Using large datasets of neurons recorded simultaneously it becomes possible to study statistically significant co-ordinative relationships between single neurons' firing activity, identifying the neurons' composing cell assemblies. The concept of cell assembly was first proposed by Hebb in 1949 in his 'Organization of Behavior', where he presents cell assemblies as the 'simplest instance of a representative process (image or idea)' (Hebb, 1949). Nowadays cell assemblies are considered as functional groups of neurons able to co-ordinate their activity in a flexible and dynamic fashion, coherently with the cognitive and perceptual processes going on in a specific state (Sakurai, 1998). According to Buzsáki (2010) 'The postulated physiological goal of the cell assembly is to mobilise enough peer neurons so that their collective spiking activity can discharge a target (reader) neuron'. Within this framework each neuron, co-ordinating its activity in different assemblies, can contribute to the encoding of multiple types of information, much like a chess simul master playing on multiple boards through its influence on downstream neurons (Londei et al., 2025).

Analytical tools have also made substantial progress over time. Although cross-correlation only allowed the detection of pairwise correlations between neurons, the introduction of advanced algorithms for cell assembly detection (CAD) (for a review see Quaglio et al. (2018) and Makdah et al. (2025)) makes possible the identification of higher-order structures of correlation, revealing previously inaccessible aspects of network architecture until some decades ago. These higher-order correlations can also be used to investigate more complex interactions between brain regions.

The present study aims to explore large-scale functional interactions between neurons belonging to 71 brain areas employing CAD as an investigative tool. This approach is useful to study the co-ordination that may underlie biologically significant interarea information flows. Specifically, the analysis of pairs of neurons, besides being instructive on the network architecture, can also provide information about a possible pairwise exchange of information, with the latency of co-ordination indicating the direction of such exchange.

We also aimed to study reentrant connectivity (Edelman, 1978), also called recurrent (Lamme & Roelfsema, 2000), on a large scale. The reciprocal nature of cortico-cortical connectivity, as described by Felleman and van Essen (1991) and later expanded by Markov et al. (2014) in macaques and by Theodoni et al. (2021) in marmosets, has shaped our understanding of the hierarchical organisation of the cortex. As proposed by Edelman (1978) reentry signalling is a fundamental principle of vertebrate brain integration (Edelman, 1993) and has been implicated as a key mechanism for generating consciousness. Edelman defines reentrant processes as those that involve one localised population of excitatory neurons simultaneously both stimulating and being stimulated by another such population (Edelman & Gally, 2013). Reentry has been proposed to serve multiple functions. For example, the reentrant connectivity between the thalamus and cortex can filter which sensory signals pass to the cortex or can be important in modulating the competition among neuronal groups for attending to specific sensory stimuli or specific contents (Edelman & Gally, 2013). We investigated reentry by focusing on the analysis of loop-like triplets identified by Londei et al. (2024), that is, three-neuron motifs distributed across two areas as a possible reentering flow of information. We tested the hypothesis that neurons participating in loop-like motifs could act as major hubs of inter-regional connectivity, functionally linking to more areas than neurons in other triplets. This larger connectivity would allow them to reprocess the information exchanged with multiple areas, assigning an integrative role to these neurons. Testing this principle at the whole-brain level across broad groups of simultaneously recorded areas was ideal for evaluating whether it represents a general principle of functional connectivity. We discovered that neurons participating in loop-like motifs, rather than in other types of three-neuron motifs, serve as the main hub neurons for connectivity with other brain areas. In other words, cells with an above-average number of connections to neurons in other brain areas play a key role in orchestrating network dynamics by being embedded in loop-like motifs. By investigating the entire brain, this association may represent a general and neurobiologically relevant organising principle rather then a peculiarity of the zona incerta (ZI) processing.

## Materials and methods

### Experimental dataset and recording

For our analysis we used a public dataset of neuronal recordings (Steinmetz et al., 2019) performed in the left hemisphere of 10 mice, both males and females, possessing heterogeneous non-epileptic genotypes, as reported by the authors: 'Ai95; Vglut1-Cre (B6J.Cg-Gt(ROSA)26Sor$^{tm95.1(CAG-GCaMP6f)Hze}$/MwarJ crossed with B6; 129S-Slc17a7$^{tm1.1(cre)Hze}$/J), TetO-G6s; Camk2a-tTa (B6; DBA-Tg(tetO-GCaMP6 s)2Niell/J crossed with B6.Cg-Tg(Camk2a-tTA)1Mmay/DboJ), Snap25-G6 s (B6.Cg-Snap25$^{tm3.1Hze}$/J), Vglut1-Cre and wild-type (C57Bl6/J)' (Steinmetz et al., 2019). Steinmetz et al. (2019) used two or three Neuropixel probes, each with 384 selectable recording sites, during each recording session. The following is a brief description of the task used to illustrate which task variables can be processed by the neurons within their assemblies. The study used a two-alternative unforced choice task as described by Burgess et al. (2017). Each trial began with the mouse briefly holding the wheel followed by the appearance of two visual stimuli (Gabor patches) with varying contrasts on the right and left screens. The task required to choose the stimulus with the higher contrast (Go trials) by rotating the wheel. This action caused the stimulus selected to move from the peripheral screen towards the central screen. Differently, the animal was required not to rotate the wheel when no stimulus was presented (NoGo trials). In trials where the stimuli had equal contrast the reward was delivered in 50% of trials, regardless of the mouse's choice. When only one stimulus was presented the animal was also required to move it towards the centre of the screen using the wheel.

Each recording session included a passive version of the principal task, where the same stimuli and trial events were repeated without requiring any choice from the animal or reward delivery. Flashed visual stimuli in the form of white squares were randomly presented on a black screen in a grid of $10 \times 36$ positions to map receptive fields (Steinmetz et al., 2019). However, it should be noted that

the passive task and the flashed visual stimuli were not included in the analysis performed in the present study. Indeed, only the aforementioned two-alternative unforced choice task was used in all the analyses presented in the following sections.

## Data analysis

Recordings were collected from a total of 72 different areas and 10 mice and were divided into 39 different recording sessions for a total of around 30,000 single units recorded. However, the ZI region was excluded from the analysis because it had been published before as the first area using our approach (Londei et al., 2024). Furthermore, we selected only neurons that fired more than 100 spikes during the execution of the two-alternative unforced choice task (Burgess et al., 2017; Steinmetz et al., 2019). This resulted in only 3834 neurons being discarded (<13%). In total, we analysed 25,858 neurons. The brain regions that were recorded by Steinmetz et al. (2019) and subsequently analysed in the present study, with the exception of the ZI, are graphically represented in Fig. 1.

## Cell assembly detection

The CAD algorithm was used to identify cell assemblies in each pair of brain regions. CAD was chosen for its flexibility and ability to detect assemblies at any temporal resolution and with any temporal activation pattern. CAD was first developed by Russo and Durstewitz (2017) and has been further optimised in its most recent version, *CADopti* (Oettl et al., 2020). Several algorithms are currently available for detecting cell assemblies (Makdah et al., 2025; Quaglio et al., 2018). However, for our purposes, *CADopti* was deemed the most suitable. We applied this algorithm to the segment of the recording covering the two-alternative unforced choice task. Assembly detection allows for the investigation of neuronal coding without relying on specific task events or behavioural readouts, as noted by Hunt et al. (2021).

*CADopti* is an unsupervised statistical approach that identifies cell assemblies by detecting recurring multi-unit activity patterns in spike trains of simultaneously recorded neurons. The algorithm tests whether specific multiunit activity patterns occur more frequently than expected by chance based on the firing statistics of the composing units. The test takes into account non-stationarities, enabling the investigation of different temporal scales of coding separately. *CADopti* explores a range of user-selected temporal resolutions (bin sizes) and lags in neuronal activity. It returns both the optimal temporal resolution and activity pattern of each supra-chance assembly detected. In this study, we explored a range of bin sizes from 0.01 to 0.1 s and respective maximum lags from 1 to 19 bins. Specifically, we used the following vectors as input: BinSizes = [0.01, 0.015, 0.02, 0.025, 0.03, 0.04, 0.05, 0.06, 0.08, 0.1] s; MaxLags = [19, 12, 9, 7, 6, 4, 3, 2, 2, 1] bins. This set of inputs allows for the detection of assemblies with temporal precision ranging from a minimum of 10 ms to a maximum of 100 ms and a maximum delay of 200 ms between the consecutive activation of two neurons.

The correlated activity in assemblies can range from perfect synchronisation on the millisecond scale to sequences of activations spanning tens or hundreds of milliseconds, depending on the reference brain area and the cognitive process underlying the firing activity (Russo & Durstewitz, 2017). *CADopti* is a suitable tool for analysing parallel spike trains generated by the activity of neurons belonging to different brain areas in an unsupervised manner. This is because *CADopti* does not require *a priori* selection of a particular neuronal correlation structure for detection, but rather extracts the optimal correlation structure from the data.

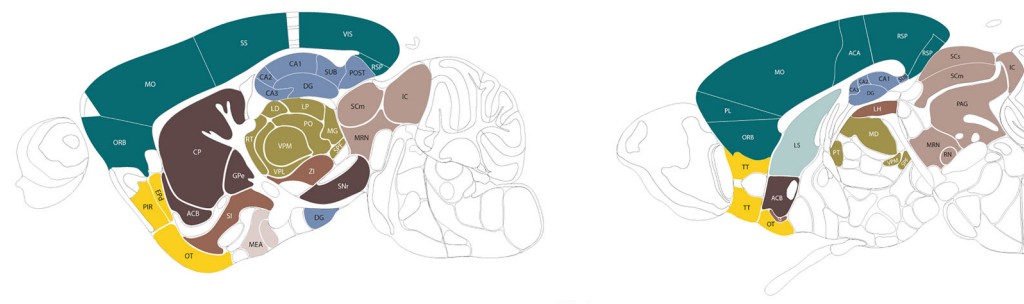

Missing areas:

DP AUD ILA MS APN NB BLA BMA COA LGd POL VAL CL

**Figure 1. Anatomical representation of recorded areas**
Anatomical distribution of areas recorded by Steinmetz et al. (2019) and integrated in our study. Colours encode the category to which each region was assigned (Fig. S1). Subdivisions of MO, SS, VIS, LS, ORB, and SC were not specified and depicted as separate. At the bottom, areas not included in the graphical representation but considered in the analysis are listed. Figures are adapted and modified from the Allen Mouse Brain Atlas. Table S1 shows the extended names of the recorded brain areas associated with each acronym.

This method consists of two main parts: a pairwise statistical test to measure the deviation of the joint spike count distribution of two neurons from the hypothesis of independence, and an agglomerative algorithm that uses this statistical test iteratively to construct assemblies with an arbitrary number of elements. The objective is to assess whether neurons' joint spike count distribution is significantly divergent from that obtained under the null hypothesis of independent processes. Therefore, after the binning of the parallel spike trains the algorithm quantifies the frequency with which a spike in one neuron is followed by a spike in the other, across a specified number of bins ($l$), for each potential lag and bin defined by the user. The bin and lag value that maximise the joint spike count are selected, with the Bonferroni–Holm procedure employed to correct for multiple comparisons. Parametric statistical testing is used to facilitate rapid computation and to account for potential non-stationarities in the time series. Subsequently, the iterative loop begins, and the algorithm adds a new neuron to the assembly set formed in the previous step, whenever possible. This is done by considering the assembly set as a new unit for pairwise testing each time. The neurons that constitute the identified assemblies can only occur once within them. The iterative algorithm stops when no new neuron can be added to a previously formed assembly set. This process is repeated using all binning values specified by the user, which define the temporal precision of the coordination. These values can be analysed separately, thanks to the non-stationarity correction applied to the statistical test. For further details on the method, refer to Russo and Durstewitz (2017).

The present study adopts, for graphical purposes, the definitions we have previously introduced, which label the overall spiking activity of each neuron as '*full-spikes activity*' (FSA) and the subset of co-ordinated spikes fired contextually to the assembly formation as '*assembly-spikes activity*' (ASA) (Londei et al., 2025).

## Assembly categories

To investigate the co-ordination between pairs of simultaneously recorded brain areas the focus was placed on assemblies consisting of two or three neurons, referred to hereafter as 'pairs' and 'triplets', respectively, following the approach adopted in our previous work (Londei et al., 2024). This approach was chosen to mitigate the otherwise prohibitive computational cost associated with analysing such a large dataset, while simultaneously enhancing the interpretability of the results. Pairs were categorised into two primary types. The first type includes neurons exhibiting precise 0-lag synchronisation, hereafter referred to as 'non-directional' or 'synchronous' assemblies. The second type involves neurons that show

a delayed activation pattern, characterised by a temporal delay in the spiking activity between assembly units, referred to as 'directional' assemblies. Assemblies that consist of neuron pairs originating from distinct brain areas (e.g., first neuron $Area_i$ and second neuron $Area_j$) are termed 'inter-regional' assemblies. Inter-regional assemblies include both directional assemblies, where the first neuron in the assembly (the 'leading' neuron) consistently precedes the activation of another neuron (the 'trailing' neuron) in a different brain area, leading to a delayed activation pattern across regions, and 0-lag assemblies, which instead exhibit a synchronous activation pattern. To assess the presence of preferred directionality in the interaction between regions, we analysed the distribution of lags among neurons in inter-regional assemblies, as previously described by Londei et al. (2024). Additionally, the term 'intra-regional assemblies' refers to assemblies composed exclusively of neurons belonging to the same brain region, again including directional and synchronous assemblies.

We then identified loop-like assemblies, specifically inter-regional triplets of three neurons, characterised by delayed chains of activation, that is, delays between the activation of successive neurons and spanning two different brain regions. Let us consider $Area_i$ as the reference area. Two categories of loop-like triplets can be defined, determined by the specific order of activation: the 'direct loop-like triplets', which involve a delayed chain of neuronal activation that begins in $Area_i$, moves to a target external $Area_j$ and then returns to $Area_i$ ($Area_i \rightarrow Area_j \rightarrow Area_i$); and the 'reverse loop-like triplets', characterised by the first and last neurons of the chain being located in the external $Area_j$, with the central neuron located in $Area_i$ ($Area_j \rightarrow Area_i \rightarrow Area_j$). In cases where the reference area is $Area_j$ direct and reverse loop-like triplets would be the opposite. Among all possible triplets between areas these loop-like assembly structures are the ones that can capture feedback or recurrent interactions within the network.

For comparison with the loop-like triplets, we also identified triplets composed of neurons from two different areas that do not exhibit the loop-like structure. These other inter-regional triplets are referred to as 'non-loop-like triplets', which include configurations such as synchronous activation of at least two units or different (unidirectional) neural chains (e.g. $Area_i \rightarrow Area_i \rightarrow Area_j$, $Area_j \rightarrow Area_j \rightarrow Area_i$).

Finally, it is crucial to emphasise that although both a pair assembly of two neurons or a loop-like chain of three neurons do not inherently imply a monosynaptic connection between them, it highlights functional co-ordination between the networks the neurons belong to. For a graphical representation of the assembly categories described above, refer to Fig. 2.

## Probability formulations and Int-Ext Index

Given that a different number of neurons were recorded in different areas, the number of assemblies identified is highly dependent on the area under analysis. To facilitate comparison of the capacity of each recorded area to form assemblies both intra- and interarea the number of assemblies identified by the algorithm as significant was normalised with respect to the total number of possible pairs or triplets of that type that could be formed on the given set of elements. Throughout the study, 'probability' refers to an empirical probability or a proportion, defined over the finite set of all possible neuron pairs or triplets. More specifically, given a recording and fixed *CADopti* parameters (i.e., possible time bins and lags), probability denotes the likelihood that a randomly selected pair or triplet from all possible combinations will be classified as an assembly by *CADopti*. This probability is calculated by dividing the number of CAD-detected pairs or triplets by the total number of possible pairs or triplets.

In mathematical terms if we define $Area_i$ and $Area_j$ as two areas for which we wish to estimate the probability of the formation of an assembly, and $|Area_i|$ and $|Area_j|$ as the number of neurons within the first and second areas, respectively, we can define the following quantities:

- **Probability of forming a pair between different areas**:

$$P_{pairs}^{ext}\left(Area_i, Area_j\right) = \frac{N_{pairs}^{ext}\left(Area_i, Area_j\right)}{|Area_i| \times |Area_j|}$$

where $N_{pairs}^{ext}(Area_i, Area_j)$ is the number of inter-regional pair assemblies identified as significant by the algorithm, that is, assemblies composed of two neurons, one belonging to $Area_i$ and one belonging to $Area_j$.

- **Normalised probability difference in pair assemblies**:

$$P_{pairs}^{difference}\left(Area_i, Area_j\right)$$
$$= \frac{P_{pairs}^{ext}\left(Area_i, Area_j\right) - P_{pairs}^{ext}\left(Area_j, Area_i\right)}{P_{pairs}^{ext}\left(Area_i, Area_j\right) + P_{pairs}^{ext}\left(Area_j, Area_i\right)}$$

- **Probability of forming a pair within the same area**:

$$P_{pairs}^{int}\left(Area_i\right) = \frac{N_{pairs}^{int}\left(Area_i\right)}{\binom{|Area_i|}{2}}$$

where $N_{pairs}^{int}(Area_i)$ is the number of assemblies consisting of two neurons both belonging to the same $Area_i$, whereas $\binom{|Area_i|}{2}$ is the binomial coefficient identifying all possible pairs of elements belonging to a set of cardinality $|Area_i|$.

- **Probability of forming a loop-like triplet without a specific order**:

$$P_{loop-like}\left(Area_i, Area_j\right)$$
$$= \frac{2 \times N_{loop-like}\left(Area_i, Area_j\right)}{|Area_i| \times |Area_j| \times \left(|Area_i| + |Area_j| - 2\right)}$$

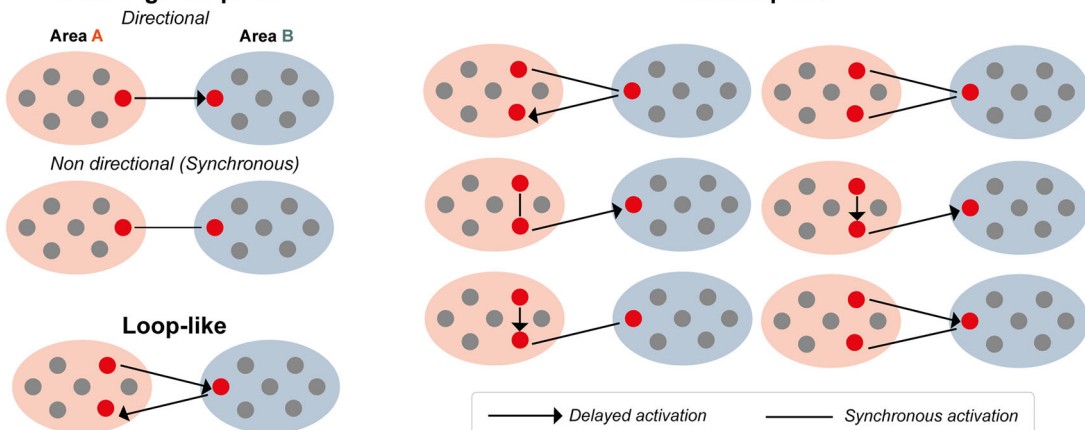

**Figure 2. Motifs and cell assembly structures**

Schematic representation of the motifs and assembly typologies used in the subsequent analysis. Inter-regional pairs characterised by assemblies consisting of two neurons spanning two distinct brain areas with delayed and directional co-ordinated activity or synchronous activity without delays (see Methods). Loop-like motif considering assemblies with three neurons, delayed co-ordinated activity spanning two distinct brain areas and with the first and last neuron of the triplet recorded in the same area (see Methods). Non-loop-like motifs include all triplets consisting of two neurons from one brain region and the other from a different brain region, characterised by the absence of a loop-like structure, that is, with synchronous co-ordination of two or three neurons or a unidirectional flow between the two distinct brain regions.

where $N_{loop-like}(Area_i, Area_j)$ is the number of loop-like triplets with first and last neuron of the chain belonging to one of the two areas and the intermediate neuron belonging to the other (note that this formula is valid for any triplet that spans two different brain areas).

- **Probability of forming a loop-like triplet with fixed chain order**:

$$P_{loop-like}^{direct}\left(Area_i, Area_j\right)$$

$$= \frac{2 \times N_{loop-like}^{direct}\left(Area_i, Area_j\right)}{|Area_i| \times |Area_j| \times (|Area_i| - 1)}$$

$$= \frac{2 \times N_{loop-like}^{reverse}\left(Area_j, Area_i\right)}{|Area_i| \times |Area_j| \times (|Area_i| - 1)}$$

$$= P_{loop-like}^{reverse}\left(Area_j, Area_i\right)$$

where $N_{loop-like}^{direct}(Area_i, Area_j)$ is the number of loop-like triplets with the first and last neuron of the chain belonging to $Area_i$ an intermediate neuron belonging to $Area_j$.

- **Normalised probability difference in loop-like assemblies**:

$$P_{loop-like}^{difference}\left(Area_i, Area_j\right)$$

$$= \frac{P_{loop-like}^{direct}\left(Area_i, Area_j\right) - P_{loop-like}^{reverse}\left(Area_i, Area_j\right)}{P_{loop-like}^{direct}\left(Area_i, Area_j\right) + P_{loop-like}^{reverse}\left(Area_i, Area_j\right)}$$

It should be noted that in instances where a particular brain region was recorded in multiple sessions, the probabilities previously outlined were determined by dividing the sum of the identified assemblies in each session by the total sum of all possible pairs or triplets of that type within each session.

In the specific case of the probability of forming loop-like structures conditioned on identified pairs (Supporting Information, sheet '*Ranking Comparison*', column '*G*') the calculation was performed using the $P_{loop-like}(Area_i, Area_j)$ formula but only the subset of neurons that form inter-regional pairs instead of the total number of neurons in $Area_i$ and $Area_j$.

Furthermore, the analysis of each area's capacity to establish functional connections with other brain areas prompted a comparison with the formation of assemblies internally, defined as assemblies consisting exclusively of neurons belonging to the same $Area_i$. To this end, we developed an index to quantify the likelihood of inter-regional assemblies, defined as assemblies between two distinct areas, $Area_i$ and $Area_j$, as opposed to the intrinsic capacity to form assemblies within the same area. This index, designated as the *Int-Ext Index* (Londei et al., 2024), quantifies the probability of inter-regional assemblies ($P_{pairs}^{ext}$) as a fraction of the sum of this probability with the probability of intra-regional assemblies ($P_{pairs}^{int}$), that is, those within $Area_i$. The index ranges from 0 to 1, with 0 representing the complete absence of inter-regional assemblies and 1 representing a state of complete absence of intra-regional assemblies. A value greater than 0.5 signifies that the probability of forming assemblies between $Area_i$ and $Area_j$ exceeds the probability of forming assemblies between neurons within the $Area_i$. Conversely, a value less than 0.5 indicates the opposite: it is more probable to form assemblies within the $Area_i$ than between $Area_i$ and $Area_j$. Finally, a value of exactly 0.5 signifies that the two probabilities are equal. Formally for a designated $Area_i$ and for each $Area_j$ recorded simultaneously with $Area_i$ we have:

- **Int-Ext Index centred on $Area_i$**:

$$I_{Int-Ext}^{Area_i}\left(Area_j\right) \equiv \frac{P_{pairs}^{ext}\left(Area_i, Area_j\right)}{P_{pairs}^{ext}\left(Area_i, Area_j\right) + P_{pairs}^{int}\left(Area_i\right)}$$

If the probability of forming both intra-regional and inter-regional assemblies was 0, the index was not calculated and an '**X**' was entered in the related Supporting Information (Sheet '*Area-Pairs Statistics*', columns '*K, L*') box instead of the index value. This occurred four times, specifically for:

$$I_{Int-Ext}^{LSr}(CP), \ I_{Int-Ext}^{LSr}(SUB), \ I_{Int-Ext}^{PL}(OLF) \text{ and}$$

$$I_{Int-Ext}^{TH}(ORBm).$$

## Hubs definition and graph-based representation

Two distinct approaches were employed to identify hub neurons, each with its own set of advantages and limitations. Generally, in the context of graph theory, a hub is defined as a node that possesses a significantly higher degree of connectivity compared to other nodes in the graph. Correspondingly, within the domain of neuroscience, a hub neuron (Bonifazi et al., 2009; Gal et al., 2021; Uzel et al., 2022) or region (Cortes et al., 2024; Sporns et al., 2007; van den Heuvel & Sporns, 2013) is characterised by its substantial number of connections, whether functional or anatomical. This heightened connectivity is deemed essential for the network's optimal functioning.

The initial approach implemented was outlined in Londei et al. (2024), wherein a threshold was established at half of the possible connections, and neurons with more than half of the potential connections were designated as hubs (referred to as the percentage method). However, this approach may have some limitations. In cases of low connectivity where no neuron has at least half of the possible connections, no neurons will be identified as hubs. Conversely, in cases of hyperconnectivity, where all neurons exceed the threshold of half the

possible connections, the identification of hubs becomes impossible. This limitation precludes statistical analysis in these specific areas. Conversely, forcibly labelling neurons as hubs in a situation with very low connectivity could lead to misinterpretation. For this reason, to confirm and validate the results, we implemented a complementary methodology based on the labelling of neurons located in the 95th percentile of the connectivity distribution as hubs (referred to as the percentile method). This strategy ensures the identification of hub neurons, thereby over-coming the potential limitation of the previous method. However, it should be noted that in cases of high connectivity, this approach may result in the exclusion of neurons that were actually hubs within their respective networks. Consequently, the findings derived from both methodological approaches are presented in this study to test whether the results obtained are consistent using different criteria.

We will consider two distinct categories of hub neurons: 'external hubs' and 'internal hubs'. External hubness is defined as the number of external areas with which a neuron co-ordinates in at least one inter-regional assembly, regardless of how many assemblies it forms with a specific external area. Conversely, internal hubness refers to the number of neurons within the same area with which a neuron forms intra-regional assemblies. In this latter case, hubness is evaluated based on the number of same-area neurons a neuron co-ordinates with, which corresponds to the number of intra-regional pair assemblies formed by that neuron. This is due to the fact that between any two neurons only a single pair assembly can exist.

To visually represent the primary finding of this study, linking the population of neurons exhibiting loop-like co-ordination patterns to hub neurons, we constructed a graph using MATLAB's native 'graph' function. This graph was centred on a specific brain region and included three categories of nodes. The first category represented external areas recorded simultaneously with the central region; these nodes were depicted with a fixed size, red colour and label with the acronym of the brain area represented. The second and third categories represented loop-like and non-loop-like neuron populations and were labelled in green and orange, respectively. The size of these nodes varied, with their diameters proportional to the number of external areas with which a specific neuron co-ordinated in at least one pair assembly. This variation in diameter reflected the neuron's external 'hubness', with larger diameters indicating higher connectivity with external brain regions.

The edges of the graph were utilised to illustrate connections in terms of assembly co-ordination. Solid red edges identified links to external areas, with one edge per external area, irrespective of the number of inter-regional pair assemblies between the neuron and neurons from that area. On the contrary, dashed light orange edges represented internal connections between neurons within the same region. The positioning of nodes was determined using MATLAB's Force-Directed Placement algorithm, as described in Londei et al. (2024). This algorithm simulated an attractive force between nodes proportional to their connectivity, resulting in more connected nodes being positioned closer together, whereas less connected nodes were placed further apart. To explicitly refer to neurons forming loop-like assemblies and representing hubs in the network we designed the expression 'embedded hubs', whereas 'not embedded hubs' refer to hubs not stemming from the loop-like population.

## Statistical tests

A binomial test was conducted to assess the statistical significance of asymmetries in detecting inter-regional pair assemblies with opposing directionality. The test quantified the occurrence of assemblies in each of the two possible directions over the total number of directional assemblies. Assuming the two directions were equally probable the null hypothesis assigned a probability of 0.5 to each direction. It is important to note that although synchronous inter-regional assembly probabilities are also presented in the results, which were calculated in the same way as delayed ones, the statistical test was conducted only on the delayed inter-regional pairs to compare the two possible directions in the flow of information. This was due to the fact that the expected probability of having a synchronous assembly depends on the specific temporal precision (bin size) at which that assembly was detected, as a different maximum lag is associated with each possible bin size. Conversely, if we restrict our analysis to only delayed pair assemblies, the probability of observing either of the two possible directions is equal under the null hypothesis of no preferred directionality.

A similar approach was adopted when assessing the statistical significance of the directional asymmetry at the level of loop-like triplets. In this case, a binomial test was adopted, albeit with a modified null hypothesis definition. Indeed within the framework of statistical analyses conducted on loop-like triplets, the null hypothesis related to the two potential chains of neuronal activation is not equiprobable. This is attributable to the feasibility of two neurons in $Area_i$ and one neuron in $Area_j$, or alternatively, one neuron in $Area_j$ and two neurons in $Area_i$. Consequently, the number of possible loop-like triplets formed in the two activation chains is contingent on the number of neurons recorded in $Area_i$ and $Area_j$. The probability of observing each of the two types of loop-like triplets has been calculated as the number of possible triplets of neurons of this type divided by the number of possible triplets of neurons in which one

neuron belongs to one area and the other two to the other area. It is important to note that this probability is specific to the particular couple of areas under consideration:

- **Expected distribution of loop-like types**:

$$P_{expected}^{direct}\left(Area_i, Area_j\right) = \frac{|Area_i|\,|Area_j|\,(|Area_i| - 1)}{|Area_i|\,|Area_j|\,(|Area_i| - 1) + |Area_j|\,|Area_i|\,(|Area_j| - 1)} = \frac{|Area_i| - 1}{|Area_i| + |Area_j| - 2}$$

For the area-pair CP-MOp, only a secondary analysis was performed to compare latencies (in terms of time bins per lag) between the two parts that composed a loop-like assembly, that is, between the activation of a CP neuron and the subsequent activation of a MOp neuron, and between the activation of an MOp neuron and the subsequent activation of a CP neuron. For this analysis, we used a one-tailed two-sample *t* test ($P < 0.05$).

Finally in comparing the proportions of hub neurons within loop-like and non-loop-like populations we opted for Fisher's exact test instead of the traditional chi-squared test. Fisher's exact test is a non-parametric approach designed to test the null hypothesis that there is no association between two categorical variables. It is particularly advantageous for small samples or datasets with skewed marginal distributions, as it avoids the distributional assumptions required by the chi-squared test. The exact *P*-value is derived directly from the sample data, making the test highly robust in these scenarios (Kim, 2017). We used a one-tailed version of the test because the question was to ascertain whether loop-like neurons exhibited a significantly higher degree of hubness compared to the non-loop-like population. In addition, we applied a correction approach to take into account the repeated application of statistical testing with Fisher's exact test by adjusting the *P*-values we obtained with the false discovery rate (FDR) correction (Benjamini & Hochberg, 1995; Trainito et al., 2019).

## Results

From the original dataset collected by Steinmetz et al. (2019), we selected and used in all the analyses reported in this paper only neurons that fired more than 100 spikes during the execution of a two-alternative unforced choice task (Burgess et al., 2017; Steinmetz et al., 2019). In total, we analysed a set of 25,858 neurons belonging to 71 different brain regions (Fig. 1) and recorded from 10 different mice in 39 different recording sessions (see Methods). Table S1 reports the full name of each recorded area and the associated acronyms used here. Supporting Information (sheet '*Area-Pairs Statistics*', columns '*B,*

*C, D*') enumerates for each area-pair the number of simultaneously recorded neurons used in our analysis and the number of mice in which such an area-pair was simultaneously recorded.

Building on our previous work (Londei et al., 2024) we investigated the inter-co-ordination of two different brain regions among all pairs of simultaneously recorded areas. Specifically, we analysed two specific motifs or patterns of co-ordination: pair assemblies and loop-like assemblies (Fig. 2). We ran the CAD algorithm (*CADopti*) on the entire dataset, excluding only those cells that did not meet the minimum number of spikes criterion described above.

We will first present the results on the functional connectivity at the pair level and at the loop-like level, after characterising the relationship between these two modalities of connectivity and showing that they represent very different interarea modalities of functional connectivity. Only at the end will we be able to integrate the results, examining the role of loop-like neurons as the main external hubs between areas in contrast to alternative triplet motifs.

### Analysis of neural pairs

A preliminary run of the *CADopti* algorithm was performed by constraining the maximum number of assembly members to two, searching for what we previously defined as pair assemblies, that is, assemblies composed of two neurons. Our research is focused on elucidating the underlying mechanisms that facilitate the interconnectivity of different brain regions in terms of mutual co-ordination. To this end, we have selected those pair assemblies identified by the algorithm that include two neurons originating from distinct brain regions, which we refer to in the Methods section as inter-regional pair assemblies. Subsequently, we divided the inter-regional pair assemblies into two further subcategories: synchronous assemblies, which are characterised by a null vector of associated lags, and directional assemblies, in which a latency exists between the activation of the first and second neuron, defined by the product of the width of the temporal bin and the value of the associated non-zero lag. For each of these categories, we estimated the probability of assembly formation for each pair of brain regions.

We then computed for each area-pair the probability of obtaining such an assembly based on the number of assemblies returned by the algorithm and the maximum number of neuron pairs that could be made based on the number of neurons recorded in each area of the pair (see Methods). For the directional cross-regional assemblies, we computed the probability for each of the two possible directions (from $Area_i$ to $Area_j$ and from $Area_j$ to $Area_i$). Each element ($i, j$) represents the probability of having a directional inter-regional pair assembly from $Area_i$ to $Area_j$. Because in general the probability of having an assembly from $Area_i$ to $Area_j$ is different from the probability of the assemblies from $Area_j$ to $Area_i$, the matrix is not symmetric. These probabilities are shown in the heatmap in Fig. 3*A*, where brain regions were sorted into groups based on a broader anatomical classification (as well as in Fig. 3*B*), as indicated in Fig. S1. Specifically we adopted, in alphabetical order, the following groups: Amygdala, Basal ganglia, Hippocampal formation, Isocortex, Midbrain, Olfactory-related areas, Septal areas and Thalamus. Eventually, we added two more groups: 'Other', collecting areas not associated with any of the anatomical groups listed above, and 'Macroarea', including areas recorded by Steinmetz et al. (2019), of which neurons had no further anatomical specification. Overall, 41.8% of the area-pairs analysed significantly differ in the probability of having an inter-regional pair assembly in one of the two possible directions (binomial test, $P < 0.05$).

In Fig. 3*B*, we report two additional important characterisations of pairs-mediated inter-regional co-ordination across the brain. The lower triangular matrix in Fig. 3*B* shows the total probability of having inter-regional pair assemblies between two areas, regardless of their directional or synchronous structure. Because this probability is commutative, that is, $P$ ($Area_i$, $Area_j$) = $P(Area_j$, $Area_i$), only the sub-diagonal submatrix is represented. This heatmap provides an immediate picture of the magnitude of functional co-ordination in pairs of areas, as well as between and within anatomical groups. For instance, it is evident that the amygdala exhibits a low probability of forming inter-regional pairs, which could indicate a lack of inter-area co-ordination, at least with the areas with which it was recorded simultaneously within this dataset. In contrast, the hippocampal formation appears to be one of the groups with the highest degree of functional co-ordination. Furthermore, the upper triangular matrix of Fig. 3*B* shows the normalised difference between the probabilities of having a directional inter-regional assembly in one of the two possible directions. Values in this matrix section that tend to −1 (shades of blue) are indicative of a predominance of directional assemblies from $Area_j$ to $Area_i$. Conversely, values that tend to 1 (shades of red) are indicative of the reverse, signifying a predominance of directional assemblies from $Area_i$ to

$Area_j$. Finally, values near zero (shades of green) indicate a similar probability in the two directions.

From Fig. 3*B* (and more detailed in Supporting Information, sheet 'Area-Pairs Statistics'), it is possible to note that the temporal framework of the co-ordination does not depend on the identity or positioning in the anatomical hierarchy of single areas involved but is rather influenced by their functional domains. For example, SCm forms directional pairs possessing only one of the two possible temporal structures when partnering with RSP but switches to originating temporally mixed pair assemblies when interacting with PL, notwithstanding the shared cortical organisation of both partners. More interestingly VISpm forms temporally mixed pair assemblies when interacting with SCig, which belongs to the motor-related domain of the superior colliculus (SC) but adopts only one assembly structure when it enters in a functional relationship with SCsg, which is part of the sensory-related domain of the SC.

Figure 3*C* and *D* show the top and bottom 30 area-pairs in terms of probability to form inter-regional assemblies. For the purposes of this analysis, only pairs of areas recorded simultaneously in at least two mice and with a minimum of 100 simultaneously recorded neurons per area were selected. However, all pairs of areas analysed are reported in the Supporting Information (sheet '*Area-Pairs Statistics*', columns '*E–H, J*') with their respective probabilities. Overall, such ranking indicates heterogeneity in the formation of inter-regional assemblies, depending on the specific groups of brain areas involved. Notably, the hippocampal formation, through a distributed contribution of its constituent areas, emerged as one of the primary structures engaged in the generation of inter-regional assemblies, mainly via intra-hippocampal co-ordination and interactions with the structure of the septal areas. Although to a lesser extent the midbrain, particularly the midbrain reticular nucleus (MRN), which co-ordinates with both other mesencephalic regions, as the SC (SCm and SCig), and with extra-mesencephalic areas, as the posterior nucleus of the thalamus (POL), also exhibited a high probability of generating inter-regional assemblies. In contrast, the amygdala, particularly the basolateral complex (BLA), exhibited limited co-ordination with other brain areas, as evidenced by its reduced probability of forming inter-regional assemblies.

Next, we explored the interplay between the detected temporal patterns of assembly co-ordination and the formation of intra- and extra-regional assemblies. To do this in a parallel analysis, we compared inter-regional assemblies, that is, assemblies encompassing neurons belonging to different areas, and intra-regional assemblies, which include only neurons populating the same area (see Supporting Information, Sheet '*Single-Area Statistics*', columns '*F–I*'). We then categorised the

assemblies into directional and synchronous to determine whether distinct co-ordinative time scales characterise inter-regional and intra-regional assemblies. Our results reveal that the distribution of probabilities for forming directional and synchronous assemblies differs across the two categories. Directional assemblies predominantly characterised external co-ordinative relationships, whereas synchronous assemblies were more commonly associated with interactions within the same area. Although common this rule did not generalise to all areas, as we observed several exceptions. PAG displayed the opposite probability distribution, with

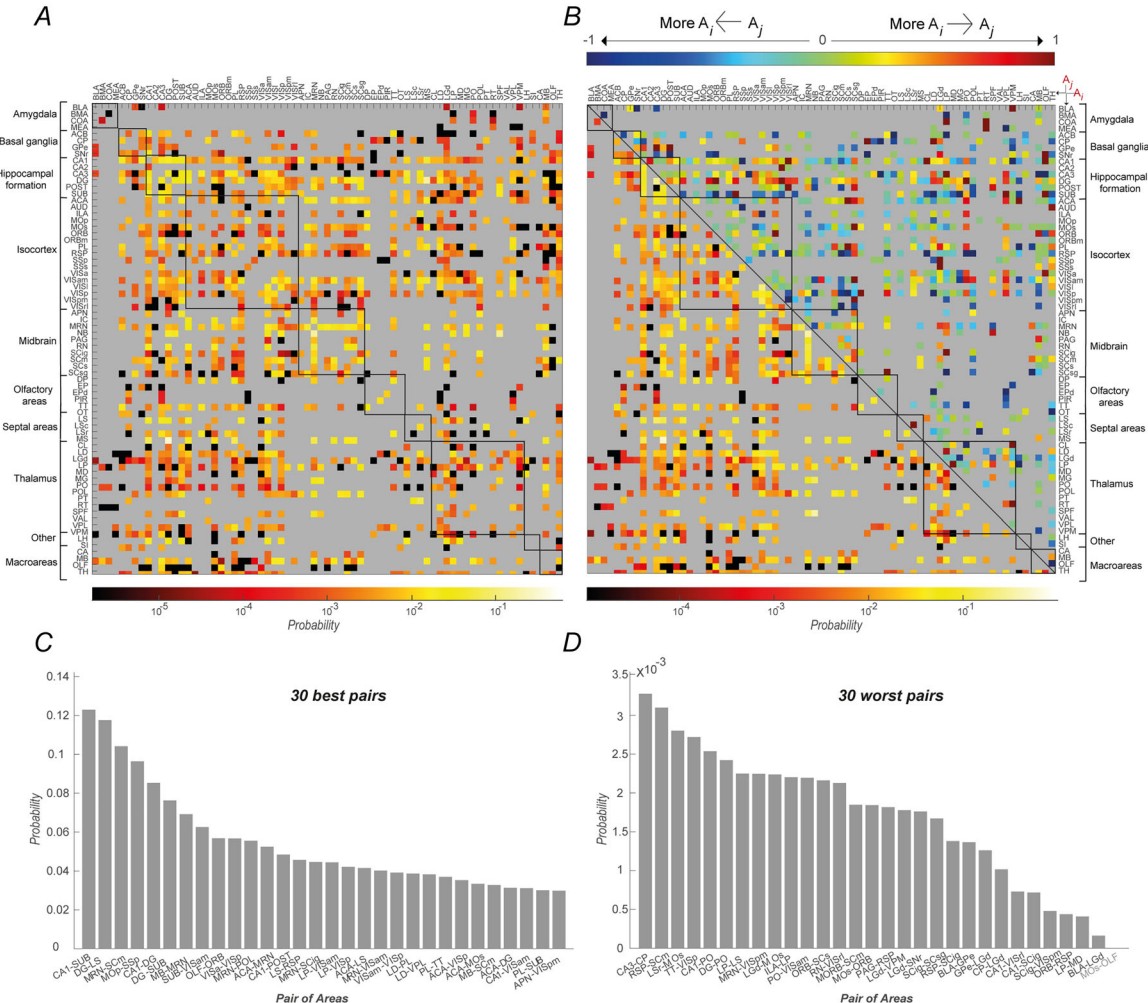

**Figure 3. Inter-regional pair assembly probabilities**

*A, B*, each pixel of the heatmaps colour-codes for the probability of the corresponding couple of areas to form inter-regional pair assemblies, considering only directional assemblies (*A*, colour bar below the heatmap), directional and non-directional assemblies (*B*, lower triangular matrix, colour bar below the heatmap) or the normalised difference between directional assemblies (*B*, upper triangular matrix, colour bar on top of the heatmap), the latter giving information on possible preferential streams of interaction. Specifically in the heatmap *A* the element located in row *i* and column *j* corresponds to the probability of forming directional inter-regional pair assemblies going from $Area_i$ to $Area_j$. Furthermore in the colour bar at the top of *B*, values closer to −1 (shades of blue) indicate a predominance of directional assemblies from $Area_j$ to $Area_i$, whereas values closer to 1 (shades of red) indicate a predominance of directional assemblies from $Area_i$ to $Area_j$. Black squares superimposed on the heatmaps enclose the areas that constitute the arbitrary groups defined for this study and reported on the *y*-axis. The black line in *B* highlights the diagonal of the heatmap. *C* and *D* represent, respectively, the couple of areas that are more and less likely to form inter-regional pair assemblies. Only areas with at least 100 neurons and recorded in at least two different mice were considered for the ranking analysis. Grey labels indicate pairs of areas in which no inter-regional assemblies were detected. Values are arranged with a logarithmic scale in *A* and *B* (lower triangular matrix), whereas a linear scale is adopted in *B* (upper triangular matrix), *C* and *D*. The light grey colour inside all area maps (*A, B*) is used to display pairs of areas that were not recorded simultaneously and therefore were not considered in the analyses, as well as pairs that consider the same area.

directional assemblies prevailing internally (68.2% of the intra-regional assemblies were directional) rather than externally (only 43.9% of the inter-regional assemblies were directional). For example, cortical areas such as ILA, ORB, SSp and all the visual areas except VISp, and subcortical regions such as LD, LP and RT, showed a prevalence of synchronous assemblies. On the contrary, the cortical MOs and the subcortical MRN, GPe, MD, SNr and PO exhibited a predominance of directional assemblies in both the intra- and inter-regional assemblies categories.

Finally, to quantify the connectivity between areas in relative terms, we defined an Int-Ext Index as a measure of each region's probability of forming inter-regional assemblies relative to its probability of forming intra-regional assemblies. As shown in the Supporting Information (Sheet '*Area-Pairs Statistics*', columns '*K, L*'), we observed, as expected, a general tendency to form more intra-regional than inter-regional assemblies. However the tendency to form more inter-regional than intra-regional assemblies is detected only in specific couples of areas, among which CA1 when co-ordinating with CA3 ($I_{Int-Ext}^{CA1}$ (CA3)= 0.503), RSP when co-ordinating with dentate gyrus (DG) ($I_{Int-Ext}^{RSP}$ (DG)= 0.502) and MD when co-ordinating with LS ($I_{Int-Ext}^{MD}$ (LS)= 0.539).

### Analysis of loop-like triplets

A second iteration of the algorithm was performed to identify triplets, more complex patterns of co-ordination, composed of three neurons In particular the analysis was restricted to what we defined as *loop-like* assemblies, that is, triplets with a latency between the activation of the three neurons and spanning two areas (say *Area$_i$* and *Area$_j$*) with an *Area$_i$* → *Area$_j$* → *Area$_i$* or an *Area$_j$* → *Area$_i$* → *Area$_j$* structure that we define as direct and reverse loop-like, respectively. In both cases, the first and last neurons are units of the same area. As observed by Londei et al. (2024) on the ZI this particular motif of inter-regional correlation seems to represent a signature of the integrative power of a certain brain area.

Figure 4 has the same general design as Fig. 3 but refers to the loop-like assemblies instead of the inter-regional pairs. Specifically, Fig. 4*A* shows the probability for each pair of areas of forming direct or reverse loop-like assemblies, as described in the Methods section. Each *(i, j)* element refers to the probability of observing a direct loop-like connection between *Area$_i$* and *Area$_j$*, which, by definition, equals the probability of having a reverse loop-like connection between *Area$_j$* and *Area$_i$*. Globally, 42.4% of the area-pairs analysed have a significant difference in the probability of having a loop-like assembly in one or the other of the possible structures of the chain of activation.

Figure 4*B* (lower triangular matrix) shows the total loop-like probability between two areas, regardless of the specific structure, direct or reverse, of the loop-like assemblies identified.

Figure 4*B* (upper triangular matrix) shows the normalised difference in the probabilities of forming each of the two possible structures of loop-like assemblies. Again, a value near to −1 or 1 indicates the prevalence of one of the two loop-like structures, direct or reverse, respectively. The inspection of Fig. 4*B*, and Supporting Information, sheet 'Area-Pairs Statistics', column '*P*', reveals that several areas 'plastically' re-adapt to originate only one of the two possible loop-like structures when changing the interacting partner. For example, LGd is able to form only reverse loop-like assemblies when coupling to VISp and only direct loop-like assemblies when coupling to LP. The latter together with other areas, such as ORB, PO and SCig, is similarly subjected to drastically reshape the temporal framework of co-ordination when engaging in functional interactions with different areas. Therefore, the tendency to adopt a specific co-ordinative architecture depends on the specific combination of areas considered.

Finally, Fig. 4*C* shows the top 30 positions in the ranking of pairs of areas based on their probabilities to form loop-like assemblies, whereas Fig. 4*D* shows the bottom 30 positions in the ranking. Once again, we included only areas recorded simultaneously in at least two mice and with at least 100 simultaneously recorded neurons. The Supporting Information section includes a comprehensive list of all the pairs of areas analysed, together with the corresponding probabilities (Supporting Information, sheet 'Area-Pairs Statistics', columns '*M–P*').

Given that higher-order assembly detection is contingent upon pair assemblies' prior identification and extension, a direct comparison of absolute probabilities across these categories is not informative. However, shifts in ranking position reflect distinct abilities to form each co-ordination structure. Overall, a direct comparison with Fig. 4*C, D* reveals that the hippocampal formation consistently exhibited a high probability of generating both pairs and loop-like assemblies. Interestingly, this was particularly evident when interactions involved regions outside the hippocampal formation, in particular the septal areas, whereas the probability decreased when co-ordination occurred within hippocampal subregions. Additionally, the midbrain, specifically via MRN, showed a strong tendency to form pair assemblies within its own macrostructure. However, in the context of loop-like assemblies this involvement expanded to include an increased co-ordination with a specific area outside the midbrain, that is, ACA of the isocortex. Moreover, loop-like assemblies offer the opportunity to extract information from the time scale of the reciprocal co-ordinative interactions. For instance,

the latency in the co-ordination between the elements of the part of the chain that goes from CP to MOp is significantly longer than that going from MOp to CP only in MOp→CP→MOp (one-tailed two-sample *t* test, $P = 0.0032$) rather than CP→MOp→CP (one-tailed two-sample *t* test, $P = 0.32$) loop-like assemblies.

### Pairs and loop-like comparison

Besides evaluating the ability of each pair of areas to form independently two or three-neuron assemblies, our focus was oriented towards the assessment of region-specific patterns in their generation and, more importantly, in the relationship between the potentiality to form pairs

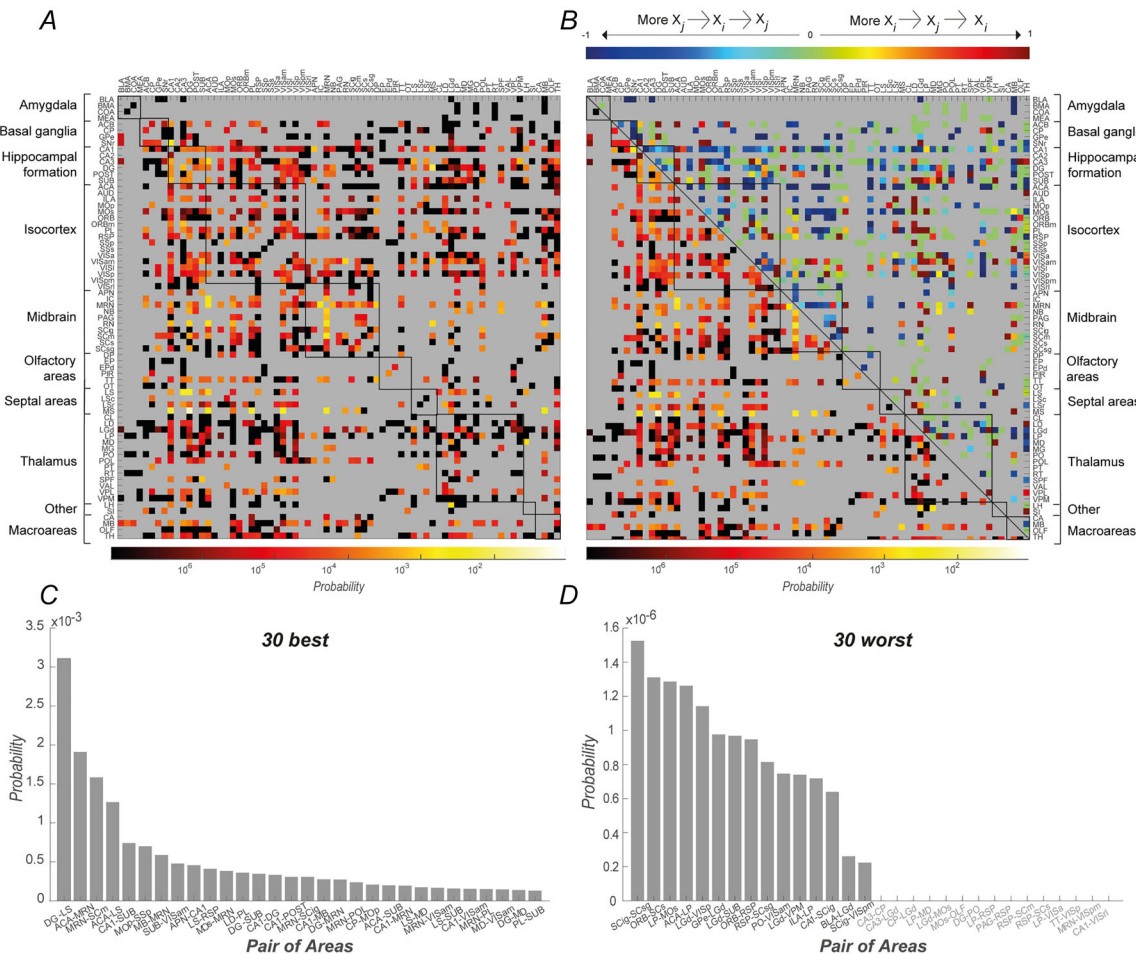

### Figure 4. Loop-like assembly probabilities

*A*, *B*, each pixel of the heatmaps colour-codes for the probability associated with loop-like assembly formation within each possible combination of areas, considering the probabilities associated with the two possible structures (*A*, colour bar below the heatmap as in Fig. 3*A*), their normalised difference (*B*, upper triangular matrix, colour bar on top of the heatmap as in Fig. 3*B*) and the probability of forming loop-like assemblies regardless of the preferred order of co-ordination (*B*, lower triangular matrix, colour bar below the heatmap as in Fig. 3*B*). Specifically in the heatmap *A* the element located in row *i* and column *j* corresponds to the probability of forming direct loop-like triplets with structure $Area_i \rightarrow Area_j \rightarrow Area_i$. Furthermore in the colour bar at the top of *B* values closer to −1 (shades of blue) indicate a predominance of reverse loop-like triplets ($Area_j \rightarrow Area_i \rightarrow Area_j$), whereas values closer to 1 (shades of red) indicate a predominance of direct loop-like triplets ($Area_i \rightarrow Area_j \rightarrow Area_i$). Black squares superimposed on the heatmaps enclose the areas that constitute the arbitrary groups defined for this study and reported on the *y*-axis. The black line in *B* highlights the diagonal of the heatmap. *C* and *D* represent, respectively, the more and less likely couple of areas to form loop-like assemblies. Only areas populated by at least 100 neurons and recorded in at least two different mice are included in the ranking analysis. Grey labels indicate pairs of areas in which no loop-like assemblies were detected. Values are arranged with a logarithmic scale in *A* and *B* (lower triangular matrix), whereas a linear scale is adopted in *B* (upper triangular matrix), *C* and *D*. The light grey colour inside all area maps (*A*, *B*) is used to display pairs of areas that were not recorded simultaneously and therefore were not considered in the analyses, as well as pairs that consider the same area.

and loop-like assemblies. Because detecting higher-order assemblies depends on the identification and subsequent extension of pair assemblies, a direct comparison of the absolute probabilities associated with the two assembly categories is not informative.

Therefore, we have decided to discuss only differences related to the position in the respective rankings. Supporting Information data (Supporting Information, Sheet *'Ranking Comparison'*) shows the positioning of each area-pair in the pair ranking and loop-like ranking, along with the absolute ranking difference. A similar ranking positioning indicates that the relative probability of forming loop-like assemblies matches that of forming pairs. In contrast, a reorganisation within the ranking positioning is indicative of a differential ability to originate the two structures of co-ordination. One extreme case of ranking reorganisation was observed between LP and two visual cortical regions, namely VISa and VISp. The combination of areas LP-VISa, in fact, ranks 32nd in the pairs ranking and 175th in the loop-like ranking, with an absolute ranking position difference of 143; similarly, LP-VISp ranks 17th and 148th, respectively, in pairs and loop-like ranking, totalling 131 as absolute ranking difference. However, when considering LP interacting with non-visual areas, such as CA1, the absolute ranking difference between the two rankings reduces to four, as pairs and loop-like probability between these area ranks, respectively, 102nd and 98th. This reorganisation in the ranking position is observed similarly in other area pairs: when partnering with ACA, RSP appears relatively better at forming loop-like assemblies (42nd) rather than pairs (109th) compared to when partnering with PAG (172nd in loop-like and 164th in pairs ranking). Further support to the specificity of forming pairs/loop-like assembly comes from analysing rankings centred on single areas, such as CP, MOp (Fig. 5*A*, *B*) and VISp (Fig. 5*H*, *I*). The comparison of pair and loop-like rankings of CP or MOp interactions with the other simultaneously recorded regions shows relatively stable and conserved rankings. At the same time, VISp displays evident rearrangements in its pattern of assemblies formed with specific areas (i.e., VISrl, VISa and LP) but not others (i.e., VISam, SCsg and ACA).

In addition to the previous comparison between rankings, we also calculated the loop-like probability with a complementary approach based on the identified pair assemblies. The formula is the same as that used for loop-like probability, but this time, instead of using all the neurons in the two areas we only used the subsets of neurons forming inter-regional pairs. However, this formulation has limitations. For example, consider the BLA-LGd pair. Despite having a low probability of forming loop-like structures, it has the highest probability of extending pairs to loop-like, even though only one loop-like structure has been

identified. This is because only two BLA neurons form assemblies with only three LGd neurons. Consequently, only nine possible loop-like structures can be formed, and the associated probability is 0.11, a very high value. This high probability has little meaning because it would risk assigning too much emphasis to a scarce co-ordination through loop-like motifs between these areas. For this reason although we have provided these probabilities in the Supporting Information (Sheet *'Ranking Comparison'*, column *'G'*), we have limited in this section the comparison to only the rankings obtained using the probabilities defined in the Methods section, $P_{pairs}^{ext}(Area_i, Area_j)$ and $P_{loop-like}(Area_i, Area_j)$.

When examining assembly-pair co-ordination between one area and the others, the pattern may be homogeneous, showing a single temporal structure in which the trailing neuron belongs in most cases to one area and the leading neuron to the other. Alternatively, the pattern may be heterogeneous, involving multiple temporal structures across its co-ordinative relationships with other regions. The same homogeneous or heterogeneous relationship between one area and the others can also occur at the level of loop-like assemblies, depending on the adherence to, respectively, one or several co-ordinative architectures. Besides being homogeneous or heterogeneous within a single assembly order, an area can show homogeneity or heterogeneity at both pair and loop-like level, revealing consistency across orders. It is important to note that when discussing the homogeneity or heterogeneity of an area, we only consider areas that form significantly more assemblies of a certain type than another, whether in terms of the direction of inter-regional delayed pair assemblies or the order of direct or reverse loop-like assemblies.

For instance, the lateral habenula (LH) is homogeneous at the pair level with the areas been recorded simultaneously with, meaning that in a statistically significant number of times the leading neuron of the assembly belonged to this area, whereas the trailing neuron belonged to an external region. This architectural homogeneity in the co-ordination is consistently maintained at the loop-like level, as it maintains the same reverse loop-like co-ordinative structures regardless of the external region it co-ordinates with (Supporting Information, sheet *'Directional Homogeneity'*).

In contrast, in the primary motor area (MOp), we found an inconsistency between pairs and loop-like levels: at the pairs level we observed a homogeneous interaction, whereas at the loop-like level a heterogeneous tendency emerged. In some cases, such as the anterior cingulate area (ACA), the anterior pretectal nucleus (APN) and the lateral septal nucleus (LS), the homogeneity is observed only at the loop-like level but not at the pair level. On the contrary, some areas, such as the CP and the SC (SCm), are consistently heterogeneous, forming assembly pairs

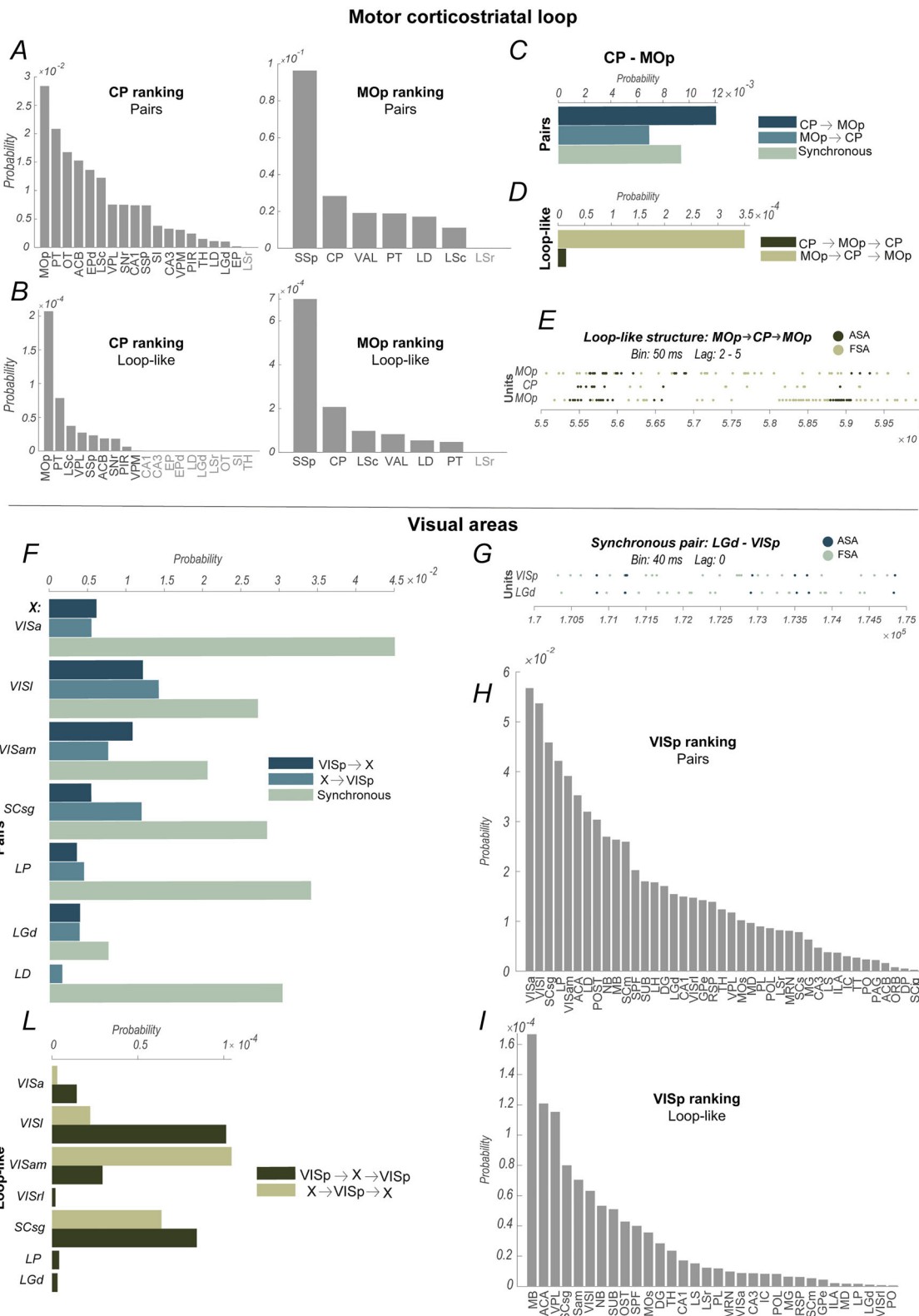

**Figure 5. Examples of pairs and triplets assembly detection in the corticostriatal and visual network**
*A–E*, assembly detection in the motor corticostriatal loop involving CP and MOp. *F–L*, assembly detection among visual-related areas. Rankings of the probability of detecting pair assemblies between (*A*) CP (left) or MOp (right) or (*H*) VISp and the areas reported on the *x*-axis. Rankings are also provided for loop-like detection between (*B*) CP and MOp or (*I*) VISp and multiple external regions. Grey region labels indicate areas in which no assemblies

were detected. The probabilities associated with the detection of assemblies having specific temporal structures are shown by multibar histograms, indicating CP-MOp distribution of (*C*) pairs and (*D*) loop-like assemblies; distribution of (*F*) pairs and (*L*) loop-like assemblies spanning VISp and multiple external regions, reported on the left. In *F*, LD has only two bars as assemblies from VISp to LD were not detected. In *L* VISrl, LP and LGd have only one bar as no reverse loop-like assemblies were identified. Examples of raster plots of pairs (*G*) and loop-like assemblies (*E*) are provided, where dark green indicates the spikes fired when the assembly is constituted (i.e., when co-ordination takes place), and light green indicates the overall firing activity.

and loop-like triplets characterised by the entire range of possible temporal structures.

An interesting case involves the co-ordination patterns expressed in and between pairs and loop-like assemblies that occurred in the dyad of CP/MOp areas, which is shown in detail in Fig. 5*C*, *D*. At the pairs level co-ordination patterns exhibited moderate heterogeneity, with a tendency to favour a temporal structure where the trailing neuron was located in the CP and the leading neuron in the MOp. In contrast loop-like assemblies drastically expanded such heterogeneity, primarily driven by the predominance of configurations in which the first and last neurons of the triplet were located in the MOp, rather than the reverse. Figure 5*E* shows an example of a loop-like assembly for CP/MOp in which the typical lagged activation patterns of such configuration can be observed at the level of spike activity associated with the assembly.

A peculiar aspect is that in some cases the homogeneity and heterogeneity seem to be shaped by the functional involvement in the same sensory domain. For instance, VISp is homogeneous at the pair level by forming synchronous assemblies only with other brain regions involved in processing visual information (Fig. 5*F*) but not with non-visual areas (Supporting Information, '*Area-Pairs Statistics*', column '*J*'). Figure 5*G* illustrates a pair assembly between LGd and VISp, with synchronous co-ordinate firing activity. Heterogeneous temporal structures of co-ordination characterise VISp loop-like level of interaction with non-visual areas (Fig. 5*L*).

### Hub neurons and loop-like triplets

In a previous study (Londei et al., 2024), we observed that neurons in the ZI form loop-like triplets, but not non-loop-like triplets, had a strong probability of forming assemblies with neurons from other brain areas. One main aim of the present study was to test whether what we defined as 'embedded hubness', that is, the association between hub neurons and loop-like motifs of interarea interaction, represents a general brain principle or is specific to the ZI modality of information processing.

We asked whether both internal and external 'hubness' were associated with the participation of neurons in loop-like motifs. Specifically, we identified and labelled as 'external hubs' neurons forming a higher-than-average

number of inter-regional pairs with neurons belonging to different regions, and as 'internal hubs' the cells with an exceptional number of co-ordinative relationships with neurons belonging to their same brain area (see Methods section for further details). Our results show that external hubs are more strongly and consistently associated with loop-like assemblies, acquiring the embedded identity with a higher probability than internal hubs, which show a labile and somehow discontinuous association with loop-like assemblies, depending on the brain area examined.

Figure 6*A* and *B* each presents two histograms illustrating the normalised proportions of embedded hubs, both for external and internal hubs, using for the hub identification the percentile (Fig. 6*A*) and percentage (Fig. 6*B*) methods, respectively (see Methods). Although external hubs were predominantly embedded hubs, as they invariably exhibited a higher proportion of hub neurons in the loop-like population, internal hubs showed only a tendency to have a higher proportion of hubs in the loop-like population. However, this is not a general phenomenon, as in some cases, an increased proportion of hubs was observed in the non-loop-like population. Indeed although all the statistical tests (Fisher's exact test, $P < 0.05$, FDR corrected) performed in the case of external hubs were highly significant considering all the 38 areas analysed and both the definitions of hub described in the Methods section, the analysis performed in the case of internal hubs yielded significant results only for 12 out of 31 (7 areas had no hubs) using the percentage method, and only for 20 out of 38 using the percentile method (Supporting Information, sheet '*Hub Statistics*'). For instance, POST, SSp and VISa not only were not statistically significant but also appeared to exhibit an opposing tendency, with the majority of hub neurons belonging to the non-loop-like population according to both the percentage and percentile methods. Note that in this analysis, we only used areas with at least 35 loop-like neurons and 35 non-loop-like neurons (about a third of the threshold of 100 neurons used in the previous analysis, as each category represents one of the three parts of a partition on the set of neurons) and recorded in at least two mice.

To graphically represent these findings, we made an undirected graph (Fig. 6*C*). In this graph centred on a specific brain area (in the case of Fig. 6*C*, DG) we distinguished three types of nodes by colour. Red

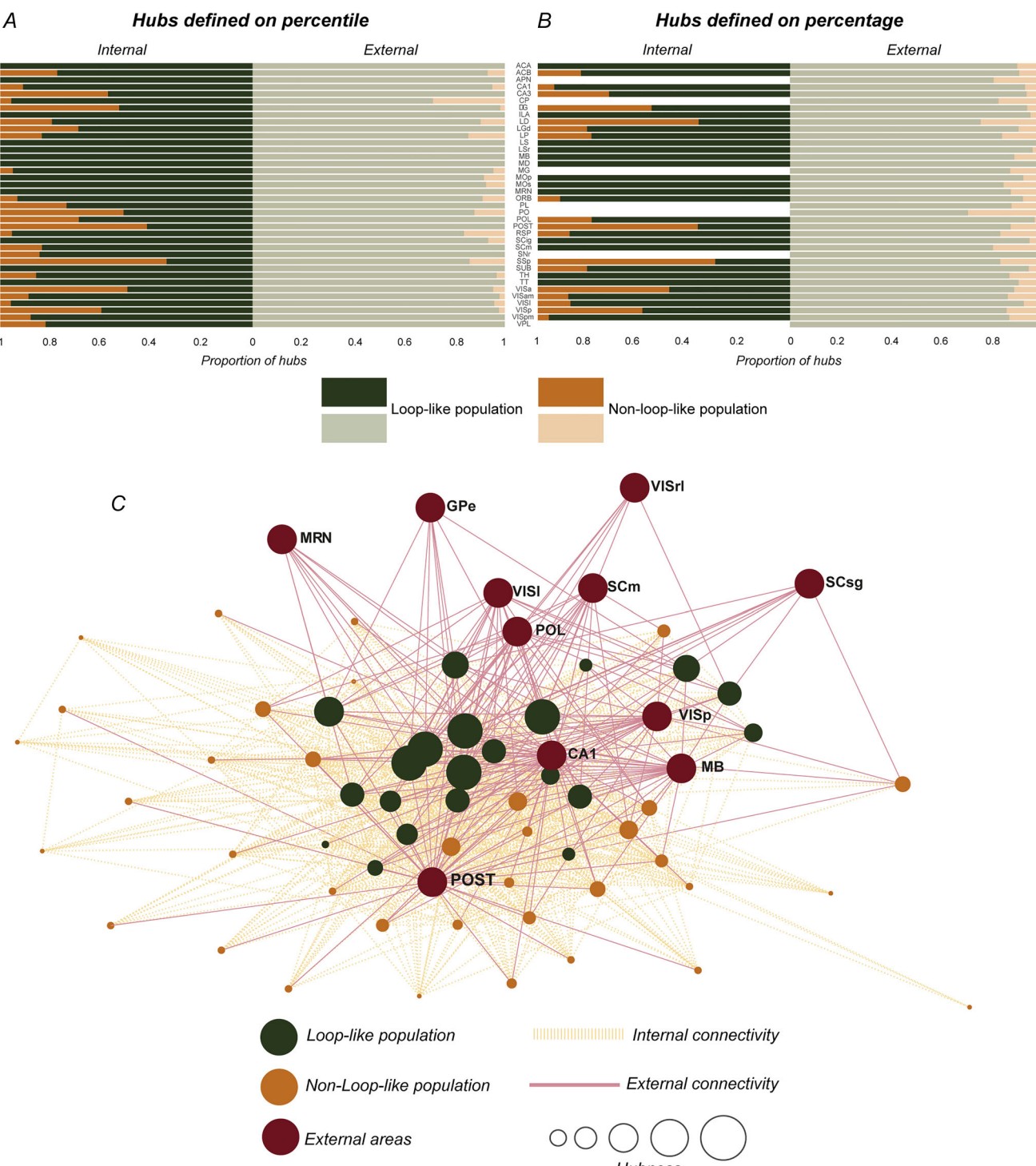

**Figure 6. Relationship between hub neurons and loop-like triplets**

*A*, *B*, histograms showing the distribution of hub neurons that form loop-like triplets (dark green on the left side of each histogram and light green on the right side of each histogram) and those that form only other types of inter-regional triplets (dark orange on the left side of each histogram and light orange on the right side of each histogram). This information is presented for both internal hubs (left histograms) and external hubs (right histograms). *C*, the undirected graph centred on a particular recording session and on one of the areas simultaneously recorded there, specifically the dentate gyrus (DG). Red nodes represent the other brain areas recorded along with DG in that session, whereas green and orange nodes represent neurons belonging to the

loop-like population (embedded hubs) and neurons part of the non-loop-like population (not embedded hubs), respectively. Continuous red edges represent connections between DG neurons and neurons from other areas, that is, if a DG neuron forms at least one pair with a neuron from another area, then such an edge is placed between that area and the DG neuron. On the contrary, orange dashed edges represent internal connections between DG neurons, that is, if a pair assembly exists between two DG neurons, then an edge of this type will exist between the two nodes representing those two neurons.

nodes represent external brain regions, whereas the nodes corresponding to neurons within the DG area are divided into two groups: neurons forming at least one loop-like triplet (green) and neurons forming only other types of inter-regional triplets (orange). The edges of the graph represent connections, defined as the presence of a pair assembly. Two types of edges connect the graph nodes: solid red edges represent connections between DG neurons and external areas, whereas dashed orange edges represent internal connections between neurons within DG. The size of the nodes representing DG neurons (orange and green nodes) is proportional to the number of external areas with which they co-ordinate in pair assemblies. This scaling reflects the 'hubness' of individual neurons, where nodes representing neurons connected to a larger number of external areas are depicted with greater diameters. At first glance, loop-like neurons (green nodes) exhibited a much higher hubness than non-loop-like neurons, despite all being part of inter-regional structures. Non-loop-like neurons formed relatively few inter-regional pairs but were not necessarily less internally connected. In contrast, non-loop-like neurons exhibited a pronounced internal microcircuitry, as evidenced by the dense orange web in the lower portion of Fig. 6*C*.

This example reflects the results obtained when comparing internal and external hub neurons, highlighting the distinct roles these two populations play in local and external network dynamics. Only the loop-like neurons show broad co-ordination with external areas. Indeed, it is important to note that this does not necessarily imply that these neurons are also hyperconnected internally, with neurons of the same brain region, as observed in DG, for example. Although the correlation between external hubs and loop-like triplets is a biological phenomenon observed across all brain regions, the correlation between internal hubs and loop-like triplets, which is also present with an overall tendency, remains a regionally specific phenomenon related to the specific brain region under consideration.

As a control, we used non-loop-like triplets of neurons. The decision to compare the loop-like population with this specific control population was based on the fact that both categories consist of triplets of neurons originating from two distinct brain areas, thus ensuring no bias in counting the external areas with which a neuron forms assemblies. However for completeness we decided to

also replicate the analysis by including all the remaining neurons together with non-loop-like triplets of neurons. Considering this larger control population of remaining neurons, which are those that were not loop-like, we found similar results to what was observed in the non-loop-like population, as illustrated in Fig. S2. This figure contains two histograms analogous to those in Fig. 6, which are based on the two definitions of hub neurons, but in this case, we compared the loop-like population with all the remaining neurons (Fig. S2A, B), which also includes the non-loop-like population. The results obtained from this analysis are comparable, with all areas exhibiting a significantly higher number of embedded hubs when considering external hubs. However, this was not always the case for internal hubs, as the results varied depending on the area considered. A graphical representation of these findings, focusing on the dorsal lateral geniculate complex (LGd), is provided in the subsequent section (Fig. S2C), where we represented all three distinct categories of neurons: loop-like neurons, non-loop-like neurons and 'other neurons', which are the unlabelled neurons. This third category of neurons is represented by the blue dots and tends to have a low level of hubness, as in the non-loop-like population.

Finally, Fig. S3 presents two additional examples of graphs, one centred on the primary visual cortex (VISp, Fig. S3A) and the other centred on the nucleus accumbens (ACB, Fig. S3B). By providing a glimpse of hub/loop-like distribution in neocortical and subcortical areas these examples complement the result obtained on the DG (Fig. 6) and show that the solid entanglement between loop-like motifs and hub neurons is conserved among evolutionarily distant and anatomically diverse structures, supporting the universality of this organising principle.

It should be noted that the result does not depend on a greater number of loop-like neurons compared to non-loop-like neurons. In fact when we added up the number of neurons in the loop-like population and compared it with the sum of the non-loop-like neurons in the 38 analysed areas, we found that the loop-like population was only 8.5% larger. Furthermore, the different sizes of the two subpopulations were considered in the statistical analysis.

For enhanced readability, the absolute numbers of hubs (calculated using the percentage and percentile methods), as well as the numbers of loop-like and non-loop-like neurons, are reported in the Supporting Information (sheet '*Hub Statistics*', columns '*K–P*').

## Discussion

In the present study, we applied a method based on the identification of cell assemblies spanning pairs of brain areas to reveal a principle of functional co-ordination that may be central to the integration of information across distant regions of the brain. Our main result is that loop-like assemblies, specific motifs of inter-regional co-ordination that implement a reentrant flow of information, constitute the primary external hubs of a given area. Remarkably, this relationship was consistently observed across all examined pairs of cortical and subcortical areas and supports the hypothesis that this co-ordinative architecture plays a key role in brain-wide information integration. Second, we found that the consistently higher external hubness of neurons participating in loop-like assemblies was not accompanied by a corresponding increase in internal hubness. Finally, we showed that loop-like assemblies do not simply reflect reciprocal connectivity at the level of neuronal pairs, and that asymmetric loop-like motifs, reentering predominantly into one area, do not necessarily correspond to analogous asymmetries at the level of neuronal pairs.

### Pairs and loop-like assemblies

Co-ordination is a long-standing, multifaceted topic in the scientific community. Over time, it has been studied from different perspectives, spanning from the inter-brain co-ordination (Kurihara et al., 2022; Mu et al., 2017) to the interarea (Schneider et al., 2021) and, lastly, interneuronal co-ordination (Londei et al., 2025; Mione et al., 2019; Nougaret & Genovesio, 2018). In terms of coding properties Londei et al. (2025) have also shown that filtering only the co-ordinated activity of neurons in the assemblies it emerges that neurons can contribute to the encoding of multiple information through their co-ordinated activity. The meaning of brain co-ordination still remains hard to fully understand. One interpretation of co-ordinative relationships relates to the 'communication through coherence' theory, which proposes that coherent and synchronised activity of different neuronal networks promotes precise and effective communication between them (Akam & Kullmann, 2012; Fries, 2015).

We investigated the extended profile of co-ordination among 71 brain areas with a cell assemblies-based strategy, which allows the search for patterned functional relationships with magnified temporal and spatial resolution. We focused on two main categories of assemblies: pairs, consisting of two units, and loop-like, three-unit assemblies spanning two areas and possessing co-ordinative delays. Our results show a highly heterogeneous distribution in the probabilities of forming these assembly categories. The pair assemblies

investigation allows the identification of a functional group of several well-known anatomical circuits. For instance the retrosplenial cortex (RSP), which is seen establishing a functional bond with several divisions of the SC, branches its projections providing a dense innervation to a large portion of this structure (García del Caño et al., 2000). An additional functional partnership, the one involving the prelimbic cortex (PL) and the SC, appears to be backed up by a strong anatomical projection arising from the former and targeting the latter (Benavidez et al., 2021). Both the RSP and the PL are functionally and anatomically related to the hippocampal formation, specifically to the field CA1, that sends monosynaptic projections to those cortical areas (Santos et al., 2023; Wyass & van Groen, 1992). Interestingly inputs from the CA1 seem to reach the visual areas either directly or indirectly through light direct connections to primary or secondary visual cortical areas, and indirectly through the projection it sends to the RSP (Cenquizca & Swanson, 2007), which, in turn, has been shown to exchange bidirectional connections with the same territories (Vann et al., 2009; Vogt & Miller, 1983). Interestingly, CAD captures both these architectures, showing assemblies spanning CA1 and primary and secondary visual areas (VISp, VISa, VISam, VISl, VISrl, VISpm) and also the RSP, besides the ones spanning the RSP and visual areas. Moreover, an extensively studied anatomical projection from the lateral geniculate nucleus (LGd) to the visual areas (Bienkowski et al., 2019) is widely represented in pairs distribution, where both the primary (VISp) and secondary (VISam and VISa) visual regions are involved in assemblies partnering with the LGd.

More importantly, the ranking of area-pairs shifts considerably depending on whether pairs or loop-like assemblies are considered. A first peculiar aspect consists in the tendency of a given brain area to interact with other brain regions following a specific pattern in the directionality of the co-ordination, with such pattern being independently determined for the pair and loop-like level of interaction.

We found that having a high probability of forming pair assemblies does not necessarily imply having a corresponding high probability of forming loop-like assemblies. Although loop-like assemblies cannot exist without pair assemblies (because of the algorithmic necessity of extending a pair to detect a triplet), we observed a very weak relationship between the probabilities of forming both. This is because the loop-like assemblies represent a specific subset within the broader category of triplets. Therefore, even when the number of triplets is high, the number of loop-like assemblies may be negligible. Conversely, even in scenarios with a limited number of triplets the proportion of loop-like assemblies may be significant, thereby increasing the probability of observing this particular type of assembly.

For instance the lateral posterior (LP) nucleus of the thalamus, which is connected to several key areas of the visual system (Allen et al., 2016; Juavinett et al., 2020) and involved in several aspects of visual processing (Aton, 2021; Cortes et al., 2024; Wei et al., 2015), modulates its co-ordination patterns depending on the specific interacting partner area (Figs 3*A* and 4*A*). When LP co-ordinates with visual-related areas, such as the anterior and primary visual area (VISa and VISp, respectively), the assembly pair is the most probable interaction mode, whereas the probability of forming loop-like assemblies is, respectively, zero or extremely low. However when co-ordinating with non-visual areas, such as CA1, not only the discrepancy between the probability of forming pair and loop-like assemblies diminishes, but the position of the interacting areas in the loop-like ranking is higher compared to the pairs ranking, suggesting that loop-like triplets do not necessarily emerge from strong functional connectivity between two areas in terms of assembly pairs, but rather may reflect an oriented pattern of co-ordination (Supporting Information, sheet '*Ranking Comparison*'). This evidence suggests that different registers of co-ordination may underlie the peculiarities of interactions between pairs of brain areas in how information is processed and, potentially, exchanged.

It is now well established the involvement of the primary motor cortex (MOp) and the caudoputamen (CP) in a motor cortico-striatal loop engaged during motor execution and learning (Cui et al., 2013; Foster et al., 2021; Marsden, 1982; Rocha et al., 2023; Roth & Ding, 2024). The interplay between MOp and striatum has been shown in several studies; for example Liu et al. (2023) showed that MOp works with the striatum for suppressing inappropriate responses, whereas Rothwell et al. (2015) reported its involvement in serial order tasks. Interestingly, the functional nature of this anatomical circuit mirrors the distribution of the loop-like assemblies formed between these two structures. The CP and the MOp reciprocally stand as the most functionally linked areas, ranking in the top positions of both rankings centred on the two areas (Fig. 5*A*), and also when considering higher-order co-ordinative structures. When we consider the loop-like motifs we found (Fig. 5*B*) that the asymmetries in distribution between MOp→CP and CP→MOp were less pronounced than those at the loop-like level, where the MOp→CP→MOp prevailed over CP→MOp→CP (Fig. 5*D* and *E* for an exemplificative raster showing the most probable temporal architecture of MOp/CP co-ordination at the triplet level). The asymmetries in distribution between MOp→CP and CP→MOp were less pronounced. The prevalence of loop-like motifs reentering the MOp is consistent with the established function of the cortico–basal ganglia–thalamo–cortical loop, which is to modulate cortical activity by enhancing appropriate

behaviours and suppressing unwanted ones through a process of disinhibition of cortical assemblies (Graybiel, 1998; Ponzi & Wickens, 2010). We were able to detect loop-like motifs even when multiple areas were interposed between the CP and the cortex. Indeed, loop-like motifs identified between two brain areas may include only a subset of the neuronal population participating in a larger assembly that can encompass multiple neurons within and across brain areas. Speculatively the longer latency of the second half of the chain in MOp→CP→MOp, rather than in CP→MOp→CP, adheres to the anatomical and functional divergence of information flow that branches downstream the CP, which, depending on the funnelling through the direct or indirect pathway, encompasses several in-between structures.

The nature and role of the anatomical intermediates between two distant populations can shape their functional interactions. Modelling studies have shown that long-range zero lag synchronisation emerges when the interaction between two populations is mediated by a third population relaying their activity (Vicente et al., 2008). Our analysis shows that areas involved in visual processing share a predominant proportion of synchronous assembly pairs (Fig. 5*F*,*G*). More specifically the primary visual area (VISp) shows the highest probability of forming assemblies with two other cortical areas involved in visual processing, which are the lateral (VISl) and the anterior (VISa) visual areas (Fig. 5*H*), with the high proportion of synchronous assemblies, rather than directional assemblies, accounting for such a close functional relationship (Fig. 5*F*). This pattern also holds true for the other cortical areas, such as VISam, VISrl, VISpm, VISa and VISl, which, similar to VISp, have preferential synchronous co-ordinative relationships with visual cortical regions (Supporting Information, sheet '*Area-Pairs Statistics*', columns '*F, G, J*'). These observations are consistent with and support the relaying hypothesis proposed by Vicente and colleagues (Vicente et al., 2008), as the aforementioned cortical regions are functionally and anatomically connected to the thalamus by cortico-thalamic loops involving several relaying thalamic nuclei (Juavinett et al., 2020). Accordingly, the high proportion of synchronous assemblies found between visual cortical and subcortical areas (VISp and SCsg) and between visual-related subcortical regions (LP and LD) may suggest the presence of neuronal populations relaying their activities. This interpretation is also plausible from a neuroanatomical perspective, as VISp and SCsg are bidirectionally connected to the LGd, whereas LP and LD share bidirectional connections with VISp (Bota, 2008).

Interestingly, although at the pair level visual areas share a trending co-ordinative structure, that is the synchronous one, such homogeneity does not pertain to the loop-like level, where the preferred co-ordinative

architecture is determined independently by the specific combination of visual areas considered. Therefore, these different co-ordination patterns may capture different aspects of the functional interaction happening within a network. Speculatively, one might hypothesise that the pair assemblies distribution is more directly influenced by the mechanistic and structural organisation of the network (i.e., the existence of relay populations), whereas the distribution of higher-order functional co-ordinative architectures, as the loop-like assemblies, may be shaped by the specific way in which the elements of the network interact.

Loop-like assemblies may represent the functional fingerprint of a particular type of between-level reentrant or recurrent activity, not necessarily linked to mono-synaptic anatomical connectivity. Reentry signalling has been implicated in several cognitive and sensory processes, including the suppression of alternative plans, the generation of coherent percepts by binding multiple sensory features, the maintenance of persistent patterns of activity and the support of attentional focus (Edelman & Gally, 2013). Loop-like co-ordination could also originate from several network mechanisms, including shared entrainment on a common oscillation, where two consecutive oscillation cycles may be associated with the activation of the first and last units of the triplet and the middle unit could be phase-locked to the same oscillation at a different phase (Domanski et al., 2023). However, their strong association across all areas with hub properties makes it unlikely.

## Motifs and hubs

Identifying the computational units of a complex network of cells interacting with each other has engaged the attention and sparked the interest of neuroscientists. The investigation into the concept of network motifs has provided a valid tool to shine light on this issue. Neurobiological investigation of network motifs has unfolded in two directions: the structural and functional levels (Milo et al., 2002, 2004; Sporns & Kötter, 2004). Structural motifs describe patterned anatomical connections that link the biological units composing a network. The same anatomical substrate may give rise to many functional motifs, defined as operative modes of inter-action between the nodes of the network (Sporns & Kötter, 2004). Interestingly, motifs transcend the neurobiological boundaries of neuron-to-neuron interactions, as they have been identified as universal design principles within several biological and non-biological systems, from food webs and transcriptional regulatory cascades to electronic circuits and the World Wide Web (Milo et al., 2002). Notwithstanding the widespread diffusion of such modular operative and structural organisation several

superfamilies of networks sharing similar key motifs have been identified, which also may have converged during evolution to perform analogous tasks (Milo et al., 2004).

The present study focuses on the analysis of triplet motifs on the basis of the attention they earned in the scientific landscape (Dechery & MacLean, 2018; Duclos et al., 2021; Gal et al., 2021; Milo et al., 2002, 2004; Ruach et al., 2023; Ye et al., 2024) and in regard to their cardinal mathematical and theoretical identity (Deshpande et al., 2023). Previous neuroscience studies have examined connectivity motifs in different species, at different scales and using different methods (Lin et al., 2024; Perin et al., 2011; Song et al., 2005). Both Song et al. (2005) and Perin et al. (2011) described a non-random organisation of neuronal assemblies in mouse slices. Song et al. (2005) performed whole-cell recordings from layer five pyramidal neurons in the rat visual cortex, showing that bidirectional connections between pairs of neurons are more common than expected by chance and that their synaptic strengths are higher than those of unidirectional connections. Moreover, they found that highly clustered three-neuron connectivity motifs were overrepresented, although it is not known whether these motifs correspond to hubs. In *Drosophila melanogaster* Lin et al. (2024) used high-resolution electron microscopy to examine three neuron motifs, not in terms of functional connectivity but at synapse-level resolution, identifying the prevalent motifs in each neuropil. However, they did not investigate the relationship between motif type and hubness. At the macroscale human connectome studies have identified areas working as major hubs and have shown that such hub regions are strongly interconnected, forming a 'rich club' that acts as an integrated core rather than as isolated nodes (Heuvel & Sporns, 2011). However, human connectome studies lack the single-cell resolution and cannot detect single-cells motifs.

Notably, our investigation relocates the search for motif-based co-ordinative relationships between neurons to the interarea level of brain-wide analysis, revealing details on how pairs of areas may route, exchange and, possibly, jointly process information. In this scenario, loop-like assemblies stand out as prominent interacting modes as they embody the functional architecture of a reentrant type of signalling between areas, which has also been hypothesised to be one of the main mechanisms through which the brain accomplishes integration (Edelman & Gally, 2013).

Integration and segregation represent two complementary processes that must be balanced to ensure the brain's capacity to effectively manipulate and transform information to originate adaptive behaviour (Lord et al., 2017). Although the latter requires a highly clustered topography to allow specific regions to carry out specific functions, the necessity for this information to be integrated in a holistic cognitive object requires the

existence of links between such segregated populations (Sporns, 2013). The so-called 'small world' network architecture is thought to grant such balance by presenting 'hubs', that is, nodes with a number of connections exceeding the average that play a central role in the orchestration of co-ordination between neurons possibly belonging to distant modules of the network (Bonifazi et al., 2009; Lord et al., 2017). The evaluation of these connections has traditionally relied on the analysis of correlations between neurons (Uzel et al., 2022), as well as on anatomical (Gal et al., 2021) and/or functional connectivity (Bonifazi et al., 2009). In the present study the criterion employed was based on the identification of specific types of assemblies. Our analysis considered two qualitatively different categories of hub neurons: the ones displaying high connectivity with neurons belonging to external regions, which we defined as 'external hubs', and those that show high intra-regional connectivity, defined as 'internal hubs'. Such categories respectively mirror the theoretical classification of hubs that sees the 'connector hubs' as nodes possessing high degree and high centrality (i.e. many connections with nodes belonging to a wide range of external modules) and accomplishing an integrative role and 'provincial hubs' as nodes possessing high degree and low centrality (i.e. many connections established within a module), which play a segregative role (Lord et al., 2017; Medaglia & Bassett, 2017; Sporns et al., 2007). Interestingly, our results show that the association between hubness and selective participation in a loop-like motif, which we defined as 'embedded hubness', is common to all the areas considered only for the external ones, whereas it becomes more dependent on the specific area when considering the internal hubs. This finding supports the hypothesis that internal and external hubs carry out qualitatively different functions within the cerebral network and, most importantly, that the loop-like assembly embodies a functional interaction motif linked and necessary to the inter-regional integrative function, which extends to the entire brain.

Given the widespread nature of such connectivity mechanisms throughout the brain further studies are needed to reveal their implication in the coding of task-related information involved in higher cognitive processes that are mediated by neuronal interactions within and between brain area networks, such as associative learning (Letzkus et al., 2015; Nougaret et al., 2024; Pasupathy & Miller, 2005; Suzuki, 2007), complex decision making (Benozzo et al., 2024; di Bello et al., 2025; Musall et al., 2019; Ramawat et al., 2022, 2023; Steinmetz et al., 2019), motor control (Candelori et al., 2025; Lara et al., 2018; Russo et al., 2020; Trautmann et al., 2025) and goal-directed behaviour (Falcone et al., 2022; Ferrucci et al., 2022b, 2022a, 2025; Freedman et al., 2001; Meshulam et al., 2025; Wallis et al., 2001).

Complementary efforts should also focus on mapping the engaged neuronal populations to their functional and morphological properties, such as neuronal time scales (Cirillo et al., 2018; Murray et al., 2014; Nougaret et al., 2021) and cell type (Ceccarelli et al., 2023, 2025; Musall et al., 2023; Najafi et al., 2020), which underlie integrative computations within and across brain areas.

### Limitations of the study

The current study shares the same limitations as our previous study on the ZI (Londei et al., 2024). The brain areas recorded can be highly heterogeneous, with different parts characterised by different patterns of afferent and efferent connections, and the recordings may be limited to just some parts of a brain area, and some areas are broadly defined as the CP, including distinct areas within them. Some areas were sampled less than others, which can affect the results due to limited sampling. Another limitation is that all the recordings were from the same hemisphere, restricting our investigation to the study of ipsilateral interarea co-ordination. Moreover for completeness some of the analyses reported in the Supporting Information refer to all 71 analysed areas and all 680 area-pairs recorded simultaneously, regardless of the number of mice on which the recordings were made (this information is reported in detail). It is important to note that results based on recordings from only one mouse should be interpreted with caution and validated by additional recordings from the same areas or area-pairs for future studies interested in building from our Supporting Information. For this reason, the main results reported in this study are restricted to those areas and area-pairs recorded in at least two mice and meeting the minimum number of neurons required for each specific analysis.

A further limitation arises when interpreting the nature of the identified functional interactions. It is important to recognise that the presence of a functional connection between two regions does not necessarily imply causality or a direct influence of one region on the other. The delayed activation or lag of two units in one assembly could represent several possibilities, including a direct or indirect excitatory effect of the first unit on the second, an indirect effect via double inhibition, a shared influence from a third region on both units with different time lags or phase-locking between the two units at different phases of a common oscillation. Definitive conclusions regarding the nature of the detected interaction can only be drawn through experimental manipulations in future studies. We must also consider that our findings may, to some degree, be task-dependent, and different tasks (e.g., those involving different sensory modalities) could lead to some differences in the results.

## Conclusion

The main result of this study is the identification of a general organisational principle of functional connectivity that remarkably applies to the whole brain without exception. Neurons involved in reentrant motifs, such as the loop-like connections between two areas, are the major hubs forming the backbone of information integration across brain areas. Although they represent the external hubs of connectivity, they do not always act as internal connectivity hubs. We have also shown that the functional type of connectivity expressed by loop-like motifs captures highly specific interaction patterns such as asymmetries in the area where the loop reenters that go beyond what emerges from neural pairs. Further research is needed to characterise the properties of these neurons, such as their coding functions, and to extend the analysis of this motif from triplets to the larger set of neurons composing the loop. In parallel by making available a large amount of data on the co-ordinative relationships the study also offers the community a resource to identify relevant and possibly unknown functional interactions. Lastly, our work can provide additional constraints to data-driven brain network models, such as the multiregion recurrent neural networks (Perich & Rajan, 2020) or multiscale, layer-dependent spiking networks with feedforward and feedback connections (Schmidt et al., 2018), helping to determine the building blocks of inter-regional inter-action and integration. At the same time, these models can also guide future work on connectivity motifs.

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

## Additional information

### Data availability statement

The dataset recorded by Steinmetz et al. (2019) and analysed in this study is available at: https://figshare.com/articles/steinmetz/9598406. The algorithm for cell assembly detection (*CADopti*) proposed by Russo and Durstewitz (2017) and used in this study is available at: https://github.com/DurstewitzLab/CADopti.

### Competing interests

All the authors declare that they have no competing interests.

### Author contributions

F.L., G.A., F.C. and A.G.: conception or design of the work; acquisition, analysis or interpretation of data for the work; drafting the work or revising it critically for important intellectual content; final approval of the version to be published; agreement to be accountable for all aspects of the work. L.F., F.S. and E.M.: acquisition, analysis or interpretation of data for the work; drafting the work or revising it critically for important intellectual content; final approval of the version to be published; agreement to be accountable for all aspects of the work.

### Funding

Sapienza Università di Roma (Sapienza): Aldo Genovesio, PH1181642DB714F6. This work was supported by Grant RYC2021-035061-I and Grant PID2022-141173NA-I00, funded by MICIU/AEI/10.13039/501100011033 and by the European Union NextGenerationEU/PRTR and ERDF/EU, respectively, awarded to EM.

### Acknowledgements

This work has been partially supported by the Sapienza University of Rome (Progetto H2020: PH1181642DB714F6 to A.G.). We are grateful to all the members of the research group (Nicholas Steinmetz, Peter Zatka-Haas, Matteo Carandini and Kenneth Harris) who shared the precious recorded data used in this study and to Eleonora Russo and Daniel Durstewitz for sharing *CADopti*.

Open access publishing facilitated by Universita degli Studi del Piemonte Orientale Amedeo Avogadro, as part of the Wiley - CRUI-CARE agreement.

### Keywords

assembly, connectomics, functional connectivity, hub, motifs

### Supporting information

Additional supporting information can be found online in the Supporting Information section at the end of the HTML view of the article. Supporting information files available:

**Peer Review History**
**Supplementary_Data_Materials**
**Supplementary Information**

