## [Peer Review History · The Journal of Physiology]

Neurons embedded in loop-like motifs act as central hubs for brain-wide integration

Fabrizio Londei, Giulia Arena, Lorenzo Ferrucci, Francesco Siano, Encarni Marcos, Francesco Ceccarelli, and Aldo Genovesio

DOI: 10.1113/JP289827

Corresponding author(s): Aldo Genovesio (aldo.genovesio@uniupo.it)

Review Timeline:

Submission Date:	30-Jul-2025
Editorial Decision:	18-Aug-2025
Revision Received:	08-Jan-2026
Accepted:	27-Jan-2026

Senior Editor: Richard Carson

Reviewing Editor: Bettina Schwab

Transaction Report:

Dear Dr Genovesio,

Re: JP-RP-2025-289827 **"Neurons embedded in loop-like motifs act as central hubs for brain-wide integration"** by Fabrizio Londei, Giulia Arena, Lorenzo Ferrucci, Francesco Siano, Encarni Marcos, Francesco Ceccarelli, and Aldo Genovesio

Thank you for submitting your manuscript to The Journal of Physiology. It has been assessed by a Reviewing Editor and by 2 expert referees and we are pleased to tell you that it is potentially acceptable for publication following satisfactory major revision.

LANGUAGE EDITING AND SUPPORT FOR PUBLICATION: If you would like help with English language editing, or other article preparation support, Wiley Editing Services offers expert help, including English Language Editing, as well as translation, manuscript formatting, and figure formatting at www.wileyauthors.com/eoo/preparation. You can also find resources for Preparing Your Article for general guidance about writing and preparing your manuscript at www.wileyauthors.com/eoo/prepresources.

REVISION CHECKLIST:

We look forward to receiving your revised submission.

Yours sincerely,

Richard Carson
Senior Editor
The Journal of Physiology

EDITOR COMMENTS

Reviewing Editor:

Comments to the Author:

Thank you for submitting your manuscript to The Journal of Physiology. It has been reviewed by two experts who both acknowledge its general relevance. Nevertheless, they have both raised critical concerns that all need to be addressed. In particular, I would like to stress that further consideration of the manuscript is dependent on showing the novelty of the work (see comments referee #1) and on choosing an appropriate statistical approach (see comments referee #2). Moreover, the impact on the field of physiology should be made clear.

REFeree COMMENTS

Referee #1:

Overall assessment:

The authors present a study analyzing simultaneously recorded mouse brain neuron spike trains from a previously published dataset (Steinmetz et al. 2019) using a previously published method (CADopti). In doing so, the authors often refer to their previous study Londei et al. 2024a, in which very similar, or at least closely related, work seems to have been carried out. They find that reciprocity correlates with hubness for some neurons in their dataset. Since this is not an entirely new finding (see references in major comments below), I would encourage the authors to carefully compare their results to previously published work in order to enhance the clarity and impact of the manuscript.

Still, with some adjustments as detailed below, the manuscript could be suitable for publication. In particular, the style of writing should be improved so that the whole paper becomes more streamlined and formulations are more concise. I only grasped the main result of the paper at the very end of the manuscript, which indicates that there is some room for improvement. The paper's readability would likely benefit from a major shortening effort.

Major comment:

1. The main results, although technically sound at large, read a bit like a statistical phenomenology. There are many results and observations that are mentioned but not discussed with respect to their functional relevance and comparison to other studies linking hubness and reciprocity. In (1), it is shown that reciprocity is often uncorrelated with degree and hence hubness at the single-neuron scale. In (2), it is shown that local cortical microcircuits show excess bidirectionality but not necessarily concentrated in hubs. In (3), hub-hub-connectivity is established as a robust macro-scale feature, and it is also shown that reciprocity depends on scale and species (4). The authors should discuss their findings in light of these results, especially highlighting which aspect of the hubness-reciprocity relationship in their work is novel.

2. It would be desirable to highlight the functional relevance of the detected pair and triplet patterns. As a first step,

comparing the results with known neuroanatomical knowledge (i.e. anatomical connectivity) seems indispensable.

3. I think that the manuscript could be significantly shortened. Maybe it would be beneficial for the reader to focus on the key results, and relay the rest to an SI. Currently, results in the SI appear in the main text and are discussed there, even though the corresponding figures are in the SI. There clearly is room for improvement here. One concrete example is the paragraph below 'Pairs and loop-like comparison'.

References

1. <https://doi.org/10.1038/s41586-024-07968-y>
2. <https://doi.org/10.1371/journal.pbio.0030068>
3. <https://doi.org/10.1523/JNEUROSCI.3539-11.2011>
4. <https://doi.org/10.1523/JNEUROSCI.3784-12.2013>

Minor comment:

The manuscript does not have page numbers and line numbers.

The abbreviation ZI is often used but never defined.

Comments pertaining to individual sections:

(some of them relate to major comments above)

Graphical Abstract: it does not become clear what a hub is and what the main results of the paper are. Most neurons shown in the figure seem unconnected. The loop-like motif on the right actually resembles a feedforward motif because the information only flows from the blue neuron to the two red neurons. Please consider adapting the figure for more clarity.

Introduction:

"has enabled the

sampling of the activity of up to tens of thousands of neurons simultaneously"- This seems excessive. Thousands of neurons is more realistic, please check this again.

"Organization of Behavior"- Please provide a proper reference.

'This aspect also lends

the concept of cell assembly to the network motifs-related domain of neuroscientific research,...'- This is a bit hard to understand, please consider simplifying the sentence.

'We tested

the hypothesis that neurons participating in loop-like motifs, for their potential role in reprocessing information exchanged between areas, act as major hubs of inter-regional connectivity, functionally

connecting with a larger number of areas compared to other triplets.'- Very long sentence, please consider shortening it.

Fig. 2: subfigure labeled 'Loop-like': This looks different from the 'loop' shown in the graphical abstract, and indeed more like a loop one would expect. Please clarify. Why is it not required for the loop to be closed, i.e. for another directional delayed activation between the two neurons in Area A?

Assembly categories:

'We then identified...'- What is the difference between direct loop-like triplets and reverse loop-like triplets? The designation of one area as external seems arbitrary. Please clarify.

'These functional structures suggest that loop-like assemblies may capture

feedback or recurrent interactions within the network.- This sentence seems trivial, please clarify.

Probability formulations and Int-Ext Index

'...is the number of inter-regional pairs assembly..'- Not clear what this means. Is it a number of pairs or a number of assemblies consisting of two neurons?

Statistical tests:

When these are the same as in the CADopti algorithm, they don't have to be mentioned again. Please clarify.

Figure 3: The resolution, especially of the labels, is too low. This needs to be fixed. I had problems understanding the figure content because of the low resolution.

Analysis of neural pairs:

Can the findings in this section be compared to anatomical (structural) connectivity?

Pairs and loop-like comparison:

'When analyzing the pattern of coordination in terms of assembly pairs between one area and the others, the coordination pattern can be homogeneous, sharing the same temporal structure of coordination, with the trailing neuron mostly belonging to that specific area and the leading neuron in one of the external regions or vice versa heterogeneous, if several temporal structures of coordination are adopted in its coordinative relationships with the other regions.'- This sentence is way too long. It needs to be shortened.

Hub neurons and loop-like triplets

'This example reflects the results obtained when comparing internal and external hub neurons, emphasizing these two populations' distinct roles in network dynamics.'- What are the mentioned different roles?

Fig.6: The panel description for panel C is missing. It is placed under the label for panel B.

It is not clear what is shown in panel A. Why are there negative values on the y-axis? Why are there different alpha values for the colors? Also the caption states that on right side of each histogram, data is shown for the distribution of hub neurons in populations of neurons that form loop-like triplets, but in the figure it says 'External', and 'Internal' for the left panel. This is very confusing. Please clarify.

Discussion- Pairs and Loop-like assemblies

'...reveal a significant reorganization in the positioning of area-pairs within the ranking of probabilities of forming the two types of assembly structures.'- This is a complicated sentence, please consider rephrasing it.

Conclusion

'The main result of this study is the identification of a general organizational principle of functional connectivity that remarkably applies to the whole brain without exception. Neurons involved in reentrant motifs, such as the loop-like connections between two areas, are the major hubs forming the backbone of information integration across brain areas.'- This should come much earlier. It is not made explicit in the text.

Referee #2:

The authors analyzed a public dataset of neuronal recordings from mouse brains to examine functional connectivity structures derived from the spiking activity of single neurons.

They employed the cell assembly detection (CAD) algorithm by Russo and Durstewitz (2017) to identify neuron pairs firing spikes at a certain temporal delay more frequently than expected from independent firing, and also neuron triplets firing spikes in specific temporal relations, termed loop-like and non-loop-like triplets.

The main finding of the study is twofold: first, the statistics of the types of triplets cannot be directly deduced from the statistics of pairs and depend on the brain region, and second, neurons constituting loop-like triplets are particularly likely to be hubs of the functional connectivity network.

The research question seems appealing to a broad range of researchers in the field, and the approach taken seems appropriate.

However, in my opinion, the statistical analyses performed are not thorough enough to support the conclusions presented by the authors, as described in detail below.

Major comments:

1.

Regarding the results presented in "Pairs and loop-like comparison" in the Results section, the relation between pairs and triplets could be more directly quantified if the authors would consider the triplet probabilities based on the number of identified pair assemblies, instead of the number of all possible pairs.

The authors correctly acknowledge that a direct comparison of pair- and triplet-probabilities is not informative, but that is because triplet-probabilities are defined on all possible pairs.

By defining the triplet-probability on the number of identified pair assemblies relevant to a specific type of triplet, the triplet-probability would properly measure how many triplets of that type are identified, taking into account how likely they are found conditioned by the already identified pair assemblies.

This would give a more direct and accurate quantification of the bias in the formation of a particular type of triplets, than the comparison of the probability ranks as the authors performed in the manuscript.

I would suggest revising this part of the manuscript to report the results in terms of the probabilities defined in this way.

2.

Regarding the results presented in "Hub neurons and loop-like triplets" in the Results section, the authors state that a majority of external hub neurons are from the loop-like population, but this statement does not make much sense without knowing how likely an arbitrary hub neuron belongs to the loop-like population.

Each hub neuron should belong to multiple triplet assemblies, and if at least one of those assemblies is loop-like, this neuron is grouped into the loop-like population.

Conversely, for a neuron to be grouped into the non-loop-like population, all those assemblies must be non-loop-like, which would be a much stricter condition than for being grouped into the loop-like population.

Thus, observing more neurons in the loop-like population than in the non-loop-like population would be a direct consequence of the definition of the populations.

To conclude the positive correlation between the hub-ness and loop-like-ness of a neuron, the authors need to show that there are more hub neurons from the loop-like population than what is expected from the definition of the populations. (I think that the triplet-probability based on identified pair assemblies that I explained above can be used to derive that expectation.)

Furthermore, I think an approach from the other direction would also be necessary: among the neurons in the loop-like population, how many are hub neurons? How is that for the non-loop-like population? Is there a clear difference between the populations in this respect?

Minor comments:

1.

The authors often refer to their previous study about ZI, but there is no explanation about what ZI is.

Please introduce this term properly.

2.

The authors compute many types of "probability", which is defined as the fraction of a specific type of pairs/triplets identified by the CAD algorithm among all possible pairs/triplets.

Technically, this should be termed as "density" rather than "probability".

For reporting how many pairs/triplets are identified among all possible cases, using the term "density" simply suffices, while using the term "probability" implies a certain assumption of a stochastic process about the formation (by the brain) or the measurement (by the experimenter) of the pairs/triplets, which I do not think necessary in the context of the present study.

I would suggest using "density" instead of "probability" throughout the manuscript.

Otherwise, the authors should clarify what stochastic process the "probability" is meant for.

3.

About the color map used in Fig. 3A-B and Fig. 4A-B for probability, it would be more intuitive and graphically appealing to me if the brighter colors corresponded to higher probabilities and vice versa.

4.

Also about these figures, the relation between (Area_i, Area_j) and (row, column) of the matrix needs to be explained in the caption.

5.

In page 16, line 1, "best and worst" should be rephrased to "top and bottom".

6.

In the last two paragraphs of page 16, please describe the findings with the actual values of the relevant measures, citing them from the supplementary spreadsheet.

7.

In the second paragraph of page 18, "the last 30 positions" should be rephrased to "the bottom 30 positions".

8.

In the second paragraph of page 20, "the external regions they were recorded with" should be rephrased to "the other simultaneously recorded regions".

9.

In the caption to Fig. 6, the explanation about "on the right (or left) side of each histogram" is not correct: that is inverted for the external hub histogram.

Also, "external hubs (left histogram)" is not correct: it is the right histogram, if I understand the figure correctly.

10.

In page 23, line 1, what "dyadic coordinative relationships" means is not clear.

11.

In the second paragraph of page 23, the term "embedded hub" should be explicitly defined before it is first used. (It is not clear enough from the description in the previous paragraph.)binary regression

END OF COMMENTS

EDITOR COMMENTS

Reviewing Editor:

Comments to the Author:

Thank you for submitting your manuscript to The Journal of Physiology. It has been reviewed by two experts who both acknowledge its general relevance. Nevertheless, they have both raised critical concerns that all need to be addressed. In particular, I would like to stress that further consideration of the manuscript is dependent on showing the novelty of the work (see comments referee #1) and on choosing an appropriate statistical approach (see comments referee #2). Moreover, the impact on the field of physiology should be made clear.

Response

As the first reviewer pointed out, motifs, hubs, and their extension into rich-club organization are widely studied topics across many species. These concepts are therefore not new per se. We have expanded our discussion to make clearer the novelty within this topic. To our knowledge, no previous study has linked these concepts in the way we do here. Specifically, we show that a particular triplet motif, defined as a loop-like triplet spanning two areas (with two neurons in one area and one in the other), is formed specifically by external (between areas), but not internal, hub neurons. Importantly, this connectivity motif characterizes the inter-area connectivity in a way that cannot be predicted by simple motifs of neural pairs, which are instead largely studied. This observation has broad implications, as it identifies loop-like neurons as promising targets for future investigations, including ongoing work in our laboratory, aimed at defining their roles in sensory, motor, and cognitive processing.

We have also addressed the two statistical issues raised by the second reviewer.

REFEREE COMMENTS

Referee #1:

Overall assessment:

The authors present a study analyzing simultaneously recorded mouse brain neuron spike trains from a previously published dataset (Steinmetz et al. 2019) using a previously published method (CADopti). In doing so, the authors often refer to their previous study Londei et al. 2024a, in which very similar, or at least closely related, work seems to have been carried out. They find that reciprocity correlates with hubness for some neurons in their dataset. Since this is not an entirely new finding (see references in major comments below),

I would encourage the authors to carefully compare their results to previously published work in order to enhance the clarity and impact of the manuscript.

Still, with some adjustments as detailed below, the manuscript could be suitable for publication. In particular, the style of writing should be improved so that the whole paper becomes more streamlined and formulations are more concise. I only grasped the main result of the paper at the very end of the manuscript, which indicates that there is some room for improvement. The paper's readability would likely benefit from a major shortening effort.

Major comment:

1. The main results, although technically sound at large, read a bit like a statistical phenomenology. There are many results and observations that are mentioned but not discussed with respect to their functional relevance and comparison to other studies linking hubness and reciprocity. In (1), it is shown that reciprocity is often uncorrelated with degree and hence hubness at the single-neuron scale. In (2), it is shown that local cortical microcircuits show excess bidirectionality but not necessarily concentrated in hubs. In (3), hub-hub-connectivity is established as a robust macro-scale feature, and it is also shown that reciprocity depends on scale and species (4). The authors should discuss their findings in light of these results, especially highlighting which aspect of the hubness-reciprocity relationship in their work is novel.

Response

We have incorporated two parts in the discussion related to the articles proposed by the reviewer on *Drosophila*, slice recording in mice, and human connectome. Interestingly, they address motifs and hubs, but at a level that differs from ours.

First part: "Previous studies have examined connectivity motifs in different species, at different scales, and using different methods (Song et al. 2005; Perin et al. 2011; Lin et al. 2024). Both Song et al. (2005) and Perin et al. (2011) described a non-random organization of neuronal assemblies in mouse slices. Song et al. (2005) performed whole-cell recordings from layer 5 pyramidal neurons in the rat visual cortex, showing that bidirectional connections between pairs of neurons are more common than expected by chance and that their synaptic strengths are higher than those of unidirectional connections. Moreover, they found that highly clustered three-neuron connectivity motifs were overrepresented, although it is not known whether these motifs correspond to hubs. In *Drosophila melanogaster*, Lin et al. (2024) used high-resolution electron microscopy to examine three neuron motifs, not in terms of functional connectivity but at synapse-level resolution, identifying the prevalent motifs in each neuropil. However, they did not investigate the relationship between motif type and hubness."

In contrast to slice work, we cannot determine the synaptic strength between connected neurons, nor can we resolve connections at synaptic resolution as with electron microscopy. Instead, we assess functional connectivity between neurons. Our approach complements such structural and synaptic techniques, as it captures statistically reliable influences

between neurons that are strong or consistent enough to be detected at the population level, while indirect or weaker interactions tend to remain undetected.

Second part: “At the macroscale, human connectome studies have identified areas working as major hubs and have shown that such hub regions are strongly interconnected, forming a “rich club” that acts as an integrated core rather than as isolated nodes (van den Heuvel and Sporns, 2023).”

Human connectome studies lack single-cell resolution and cannot detect single-cell motifs. An open question raised by these studies in the context of our results is whether loop-like neurons at the microscale of single cell levels show analogous rich club properties. Future studies should test whether these neurons function not only as external hubs but also display rich-club–like organization.

As in other studies characterizing hubs and motifs, we have not yet addressed their relationship with functional properties. We believe that our findings go beyond a mere statistical description of functional connectivity patterns, as they end up revealing a principle of functional coordination that applies to all pairs of areas recorded together: loop-like neurons are the external (not internal) brain hubs. As a next step, we plan to investigate the functional role of loop-like neurons for their coding properties.

2. It would be desirable to highlight the functional relevance of the detected pair and triplet patterns. As a first step, comparing the results with known neuroanatomical knowledge (i.e. anatomical connectivity) seems indispensable.

Response

We thank the reviewer for the suggestion. We have now expanded the discussion section (lines 875-897), highlighting that some relevant anatomical projections have a functional counterpart in the distribution of cell assemblies across the involved areas. However, we wish to emphasize that anatomical and functional connectivity do not necessarily unfold along the same trajectories; for instance, a strong anatomical projection might not be recruited during a particular behavioral task that cues a specific cognitive resource, resulting in a mismatch with the functional connectivity profile. This mismatch is especially evident at the level of loop-like assemblies, as discussed in the paper.

Let us consider, for example, the cases in which two areas show high functional reciprocal connectivity at the level of neural pairs, potentially reflecting strong anatomical connectivity, but do not show coordination at the level of loop-like triplets. These cases indicate that loop-like triplets do not simply reflect reciprocal connections mediated by pairs of neurons. The opposite situation is also possible, as we observed a high relative probability of forming loop-like motifs in conjunction with a low probability of forming pairs.

3. I think that the manuscript could be significantly shortened. Maybe it would be beneficial for the reader to focus on the key results, and relay the rest to an SI. Currently, results in the SI appear in the main text and are discussed there, even though the corresponding figures are in the SI. There clearly is room for improvement here. One concrete example is the paragraph below 'Pairs and loop-like comparison'.

Response

We briefly introduced the analysis pipeline at the beginning of the results section in the revised manuscript, anticipating the main result. We considered moving earlier the final part concerning the relationship between hubs and loop-like neurons; however, the main analysis builds upon the preceding analyses, and therefore their order cannot be changed without compromising the logical structure of the results.

We have now added the following paragraph to the Results section:

“We will first present the results on the functional connectivity at the pair level and at the loop-like level, after characterizing the relationship between these two modalities of connectivity and showing that they represent very different inter-area modalities of functional connectivity. Only at the end will we be able to integrate the results, examining the role of loop-like neurons as the main external hubs between areas in contrast to alternative triplet motifs.”

References

1. <https://doi.org/10.1038/s41586-024-07968-y>
2. <https://doi.org/10.1371/journal.pbio.0030068>
3. <https://doi.org/10.1523/JNEUROSCI.3539-11.2011>
4. <https://doi.org/10.1523/JNEUROSCI.3784-12.2013>

Minor comment:

The manuscript does not have page numbers and line numbers.

Response

We have now added page and line numbers to the manuscript, thanks.

The abbreviation ZI is often used but never defined.

Response

We have now defined ZI as zona incerta in the manuscript where we first introduced the term and added it to Table S1, "Clarification of Area Acronyms," in the Supplementary Information file.

Comments pertaining to individual sections:

(some of them relate to major comments above)

Graphical Abstract: it does not become clear what a hub is and what the main results of the paper are. Most neurons shown in the figure seem unconnected. The loop-like motif on the right actually resembles a feedforward motif because the information only flows from the blue neuron to the two red neurons. Please consider adapting the figure for more clarity.

Response

We thank the reviewer for pointing out the graphical error. We have corrected the arrows in the graphical abstract so that they properly represent the loop-like motif, and we have also simplified the figure and added the following caption:

“Neurons form loop-like triplet motifs in which activity originates in Area i, propagates to an external Area j, and returns to Area i. Neurons participating in these loops act as major external hubs, forming pairwise assemblies with neurons across multiple other areas (A–N). This identifies loop-like motifs as key integrative elements in large-scale cortical communication, unlike other motifs that lack hub-like properties.”

Introduction:

"has enabled the sampling of the activity of up to tens of thousands of neurons simultaneously"- This seems excessive. Thousands of neurons is more realistic, please check this again.

Response

We have adjusted the sentence, neuropixel probes work on the order of hundreds of neurons, SiNAPS on the order of 1024 per electrode, with multiple SiNAPS probes would be possible to reach few thousands:

“has enabled the sampling of the activity of up to hundreds of neurons simultaneously”

"Organization of Behavior"- Please provide a proper reference.

Response

In the submitted manuscript, we had included a reference to a 2005 reprint of Hebb's famous book in order to obtain a DOI. We have now replaced it with the reference to the original 1949 work.

'This aspect also lends the concept of cell assembly to the network motifs-related domain of neuroscientific research,...'- This is a bit hard to understand, please consider simplifying the sentence.

Response

We have reformulated this sentence:“This aspect also lends the concept of cell assembly to the network motifs-related domain of neuroscientific research, which aims at identifying elementary patterns of interactions between nodes of more complex and vast networks.” as

“These higher-order correlations can also be used to investigate more complex interactions between brain regions.”

'We tested the hypothesis that neurons participating in loop-like motifs, for their potential role in reprocessing information exchanged between areas, act as major hubs of inter-regional connectivity, functionally connecting with a larger number of areas compared to other triplets.'- Very long sentence, please consider shortening it.

Response

We have improved and split the sentence:

“We tested the hypothesis that neurons participating in loop-like motifs could act as major hubs of inter-regional connectivity, functionally linking to more areas than neurons in other triplets. This larger connectivity would allow them to reprocess the information exchanged with multiple areas, assigning an integrative role to these neurons.”

Fig. 2: subfigure labeled 'Loop-like': This looks different from the 'loop' shown in the graphical abstract, and indeed more like a loop one would expect. Please clarify. Why is it not required for the loop to be closed, i.e. for another directional delayed activation between the two neurons in Area A?

Response

We thank the reviewer for noticing the error in the graphical abstract. We have corrected and adjusted it to match the one in Fig. 2, which is the correct one.

The only reason the loop-like does not require an additional delayed activation between the two neurons in Area A is due to the structure of the algorithm: the same neuron cannot appear twice in the same assembly. This is precisely why it is called “loop-like”, nevertheless preserving a loop flow between areas, where activation starts from one area, moves to another, and finally returns to the initial area that composes the assembly.

To emphasize this feature of the algorithm, we added the following text in the Methods section dedicated to CADopti:

“It should be noted that, by design of the algorithm, the same neuron cannot appear twice in the same assembly.”

Assembly categories:

'We then identified...'- What is the difference between direct loop-like triplets and reverse loop-like triplets? The designation of one area as external seems arbitrary. Please clarify.

Response

The definition of direct and indirect refers to the reference area. For example, if area A has only loop-like with the trailing neuron in it, it would have only direct loop-like triplets.

We have made it clearer in the methods of the manuscript (the parts in bold are the new ones):

“Let us consider $Area_i$ as the reference area. Two categories of loop-like triplets can be defined, determined by the specific order of activation: the "direct loop-like triplets", which involves a delayed chain of neuronal activation that begins in $Area_i$, moves to a target external $Area_j$, and then returns to $Area_i$ ($Area_i \rightarrow Area_j \rightarrow Area_i$); and the "reverse loop-like triplets", characterized by the first and last neurons of the chain being located in the external $Area_j$, with the central neuron located in $Area_i$ ($Area_j \rightarrow Area_i \rightarrow Area_j$). **In case the reference area is $Area_j$ direct and reverse loop-like triples would be the opposite.”**

These functional structures suggest that loop-like assemblies may capture feedback or recurrent interactions within the network.- This sentence seems trivial, please clarify.

Response

We have revised the sentence to make clear that our evaluation is comparative:

“Among all possible triplets between areas, these loop-like assembly structures are the ones that can capture feedback or recurrent interactions within the network.”

Probability formulations and Int-Ext Index

'...is the number of inter-regional pairs assembly..' - Not clear what this means. Is it a number of pairs or a number of assemblies consisting of two neurons?

Response

By inter-regional pair assemblies, we mean assemblies of two neurons from two areas, thus composed of one neuron from $Area_i$ and one from $Area_j$. We have modified the sentence to make it clearer as follows:

“...is the number of inter-regional pair assemblies identified as significant by the algorithm, that is, assemblies composed of two neurons, one belonging to $Area_i$ and one belonging to $Area_j$

Statistical tests:

When these are the same as in the CADopti algorithm, they don't have to be mentioned again. Please clarify.

Response

CADopti only returns assemblies that are statistically significant under the null hypothesis of independence between the spike trains of the neurons that compose the assembly. This test is performed directly within the algorithm and corrected for multiple comparisons using Bonferroni-Holm correction. As a result, the assemblies analyzed have already passed

statistical significance checks, therefore this test is taken for granted and is not repeated in the text each time specific groups of assemblies are considered.

Figure 3: The resolution, especially of the labels, is too low. This needs to be fixed. I had problems understanding the figure content because of the low resolution.

Response

Thank you for pointing this out. The high-resolution figures to be used for publication have higher resolution.

Analysis of neural pairs:

Can the findings in this section be compared to anatomical (structural) connectivity?

Response

The reviewer highlights a common limitation of large-scale studies such as ours, that is the difficulty of addressing the implications of individual results. Our study includes 680 pairs of areas. While we addressed anatomical comparisons in great detail for the zona incerta in a previous study, such a level of detail is not feasible here. As a result, our discussion of the relationship between functional and anatomical connectivity is necessarily limited to a small number of areas. Nevertheless, we have added additional examples in the Discussion that may be useful in this context.

Pairs and loop-like comparison:

'When analyzing the pattern of coordination in terms of assembly pairs between one area and the others, the coordination pattern can be homogeneous, sharing the same temporal structure of coordination, with the trailing neuron mostly belonging to that specific area and the leading neuron in one of the external regions or vice versa heterogeneous, if several temporal structures of coordination are adopted in its coordinative relationships with the other regions.'- This sentence is way too long. It needs to be shortened.

Response

We split the sentence into two:

“When examining assembly-pair coordination between one area and the others, the pattern may be homogeneous, showing a single temporal structure in which the trailing neuron belongs in most cases to one area and the leading neuron to the other.

Alternatively, the pattern may be heterogeneous, involving multiple temporal structures across its coordinative relationships with other regions.”

Hub neurons and loop-like triplets

'This example reflects the results obtained when comparing internal and external hub neurons, emphasizing these two populations' distinct roles in network dynamics.'- What are the mentioned different roles?

Response

We made the sentence more clear:

“This example reflects the results obtained when comparing internal and external hub neurons, highlighting the distinct roles these two populations play in local and external network dynamics. Only the loop-like neurons show broad coordination with external areas.”

Fig.6: The panel description for panel C is missing. It is placed under the label for panel B.

Response

We thank the reviewer for pointing out this typo. We have modified the caption so that the letters now refer to the correct parts of Figure 6.

It is not clear what is shown in panel A. Why are there negative values on the y-axis? Why are there different alpha values for the colors? Also the caption states that on right side of each histogram, data is shown for the distribution of hub neurons in populations of neurons that form loop-like triplets, but in the figure it says 'External', and 'Internal' for the left panel. This is very confusing. Please clarify.

Response

We thank the reviewer for pointing out this typo, which was due to an earlier version of Figure 6. The negative values were a typing error. In the revised manuscript only positive values are shown. These values indicate the proportion of hub neurons belonging to the two categories analyzed (loop-like population and non-loop-like population). The different shades of color are used to differentiate between the analyses performed on internal hubs (dark colors) and those performed on external hubs (light shaded colors). We have also corrected the caption regarding the right and left sides of the histograms, which, as the reviewer correctly pointed out, refer to internal hubs and external hubs, not to loop-like and non-loop-like populations, which are identified by the colors green (dark and light shaded) and orange (dark and light shaded), respectively. For clarity, we have also included the caption with these changes below:

“Fig.6. Relationship between hub neurons and loop-like triplets. (A-B) Histograms showing the distribution of hub neurons that form loop-like triplets (dark green on the left side of each histogram and light green on the right side of each histogram) and those that form only other types of inter-regional triplets (dark orange on the left side of each histogram and light orange on the right side of each histogram). This information is presented for both internal hubs (left histograms) and external hubs (right histograms). (C) The undirected graph centered on a particular recording session and on one of the areas simultaneously recorded there, specifically the dentate gyrus, DG. Red nodes represent the other brain areas recorded along with DG in that session, while green and orange nodes represent neurons belonging to the loop-like population (embedded hubs) and neurons part of the non-loop-like population (not embedded hubs), respectively. Continuous red edges represent

connections between DG neurons and neurons from other areas, i.e., if a DG neuron forms at least one pair with a neuron from another area, then such an edge is placed between that area and the DG neuron. On the other hand, orange dashed edges represent internal connections between DG neurons, i.e., if a pair assembly exists between two DG neurons, then an edge of this type will exist between the two nodes representing those two neurons.”

Discussion- Pairs and Loop-like assemblies

'...reveal a significant reorganization in the positioning of area-pairs within the ranking of probabilities of forming the two types of assembly structures.'- This is a complicated sentence, please consider rephrasing it.

Response

We rephrased and split the sentence to make it clearer:

“Our results show a highly heterogeneous distribution in the probabilities of forming these assembly categories.” [...] “More importantly, the ranking of area pairs shifts considerably depending on whether pairs or loop-like assemblies are considered.”

Conclusion

'The main result of this study is the identification of a general organizational principle of functional connectivity that remarkably applies to the whole brain without exception. Neurons involved in reentrant motifs, such as the loop-like connections between two areas, are the major hubs forming the backbone of information integration across brain areas.'- This should come much earlier. It is not made explicit in the text.

Response

We have further developed the initial part of the discussion, highlighting the generality of the relationship between loop-like motifs and hubs:

“In the present study, we applied a method based on the identification of cell assemblies spanning pairs of brain areas to reveal a principle of functional coordination that may be central to the integration of information across distant regions of the brain. Our main result is that loop-like assemblies, specific motifs of inter-regional coordination that implement a reentrant flow of information, constitute the primary external hubs of a given area. Remarkably, this relationship was consistently observed across all examined pairs of cortical and subcortical areas and supports the hypothesis that this coordinative architecture plays a key role in brain-wide information integration. Second, we found that the consistently higher external hubness of neurons participating in loop-like assemblies was not accompanied by a corresponding increase in internal hubness. Finally, we showed that loop-like assemblies do not simply reflect reciprocal connectivity at the level of neuronal pairs, and that asymmetric loop-like motifs, reentering predominantly into one area, do not necessarily correspond to analogous asymmetries at the level of neuronal pairs.”

Referee #2:

The authors analyzed a public dataset of neuronal recordings from mouse brains to examine functional connectivity structures derived from the spiking activity of single neurons.

They employed the cell assembly detection (CAD) algorithm by Russo and Durstewitz (2017) to identify neuron pairs firing spikes at a certain temporal delay more frequently than expected from independent firing, and also neuron triplets firing spikes in specific temporal relations, termed loop-like and non-loop-like triplets.

The main finding of the study is twofold: first, the statistics of the types of triplets cannot be directly deduced from the statistics of pairs and depend on the brain region, and second, neurons constituting loop-like triplets are particularly likely to be hubs of the functional connectivity network.

The research question seems appealing to a broad range of researchers in the field, and the approach taken seems appropriate.

However, in my opinion, the statistical analyses performed are not thorough enough to support the conclusions presented by the authors, as described in detail below.

Major comments:

1.

Regarding the results presented in "Pairs and loop-like comparison" in the Results section, the relation between pairs and triplets could be more directly quantified if the authors would consider the triplet probabilities based on the number of identified pair assemblies, instead of the number of all possible pairs.

The authors correctly acknowledge that a direct comparison of pair- and triplet-probabilities is not informative, but that is because triplet-probabilities are defined on all possible pairs.

By defining the triplet-probability on the number of identified pair assemblies relevant to a specific type of triplet, the triplet-probability would properly measure how many triplets of that type are identified, taking into account how likely they are found conditioned by the already identified pair assemblies.

This would give a more direct and accurate quantification of the bias in the formation of a particular type of triplets, than the comparison of the probability ranks as the authors performed in the manuscript.

I would suggest revising this part of the manuscript to report the results in terms of the probabilities defined in this way.

Response

We thank the reviewer for raising this issue, which we had already considered. We fully agree with the reviewer that quantifying the probability of triplets based on the number of pairs returned by the algorithm that are relevant for a specific type of triplet would allow for a more direct comparison between the probabilities of forming pairs and triplets. Accordingly, we repeated the analysis on triplets, this time considering not all possible triplets that could potentially be formed from the initial group of neurons, but only those that could be formed based on the pairs returned by the algorithm. To do this, we applied two different approaches, one more approximate but which maintains the definition of probability present in the methods, and the other more accurate but that requires a more complex procedure to implement. In particular, the first approach, referred to as Subset Probability, is based on identifying the subset of neurons that form pairs with the other area. Once these subgroups of the initial set of neurons were defined, we applied the same probability formula defined in the methods but restricted it only to neurons that could actually form loop-like structures. In the second method, which we will refer as Exact Count Probability, we counted the exact number of possible loop-like structures by combining two pair assemblies with certain properties, taking into account the identity of the neurons, the directionality and synchronicity at the pair level, and the structure of the algorithm. In particular, the two pairs must have the following properties: 1) the two pairs share a neuron; 2) at least one of the two pairs is directional (i.e., with a lag other than zero; this condition is sufficient because, given the design of the algorithm, at least the directional pair can be extended to loop-like with the addition of the third neuron from the other pair, which can be either synchronous or directional); and 3) the three neurons resulting from the union of the two pairs must belong to two different areas. This required evaluating all the pairs to which each neuron belongs.

Although the new probabilities generated accurately reflect the probability of a specific type of triplet based on pairs (in a mathematical parallel, they reflect the conditional probability), as we already discussed when drafting the article, this approach also has its limitations. These limitations become apparent when we consider that ranking area pairs by the new triplet probability places the BLA-LGd pair in first place using the Subset Probability. However, the algorithm only returns one loop-like motif for this pair. In fact, with the original probability, it occupied one of the last positions. This is because, based on the small number of neurons forming pairs, only nine loop-like structures can be constructed. Therefore, the probability of finding loop-like structures under the new definition is one-ninth, which is very high. Nevertheless, considering a pair that forms only one loop-like structure as the pair with the highest probability of forming loop-like structures seems little useful to describe the coordination between areas because such a small number of loop-like structures can have a very limited role in the functional coordination between areas. What we are actually evaluating is the probability that the pairs found will become loop-like, not the absolute probability of finding loop-like structures between those two regions. In any case, the pair of areas BLA-LGd had a fairly high number of neurons (500 in total, with more than 200 in each area), so they could potentially form many loop-like structures (almost four million). However, they form only one loop-like structure, so they should generally have a very low associated probability. Using the Exact Count Probability, we observe similar results with the pair that gains 118 positions out of 163 total positions (or 178 positions, including also those with zero probability to form loop-like structures). There are several similar examples. For instance, the SCig-VISpm area pair has only one loop-like structure and a total of 476 neurons. Using the two new probabilities, it gains 135 and 76 ranking positions. Other examples include BLA-GPe, SCig-SCsg, CA1-SCig, and RSP-SCig. Based on the initial populations of

neurons, all of these area pairs have millions of potential loop-like structures that can be constructed. Even if only very few of these structures were found, however, they gained tens of positions based on the new definition of probability.

A special case is covered by the MOs-OLF pair, for which no loop-like structure has been identified. However, thanks to the new probability based on pairs, we know that this is due to the fact that no inter-regional pairs were identified between the two areas and, consequently, no loop-like can be created from the pairs. As a result, this pair has a value equal to NaN (not a number) with the new probability definitions.

Despite this, there are also many results that remain comparable across the different definitions of probability. For example, three of the top four positions linked to our original definition of probability remain in the top ten positions in the rankings of both of the new probability measures based on pairs, specifically DG-LS, ACA-MRN, and ACA-LS. Furthermore, all five examples cited in the main text remain comparable in terms of ranking comparison when compared using the new probabilities we defined. For this reason, we decided to retain the new analysis alongside the one originally proposed since, as mentioned, they reflect different aspects of the formation of the specific types of triplets analyzed.

Since we do not observe any dramatic differences between the results obtained using the two new probabilities, we have decided to present the manuscript only with the Subset Probability. This choice is motivated by its ease of reproducibility, and by our desire to avoid overloading the article with a detailed algorithmic description of how to compute possible loop-like structures.

Therefore, while maintaining the original analyses, we included values related to the newly defined probability, which estimates how likely the identified pairs might organize into specific types of triplets. These values have been added as a new column in the Supplementary Table. The following description has also been added to the Methods section, where we explain how the estimated number of loop-like structures starting from pairs was counted:

“In the specific case of the probability of forming loop-like structures conditioned on identified pairs (Supplementary Materials, sheet “*Ranking Comparison*”, column “*G*”), the calculation was performed using the $P_{loop-like}(Area_i, Area_j)$ formula but using only the subset of neurons that form interregional pairs instead of the total number of neurons in area $Area_i$ and area $Area_j$.”

The following sentences have been added to the Results section:

“In addition to the previous comparison between rankings we also calculated the loop-like probability with a complementary approach based on the identified pair assemblies. The formula is the same as that used for loop-like probability but this time, instead of using all the neurons in the two areas, we only used the subsets of neurons forming inter-regional pairs. However, this formulation has limitations. For example, consider the BLA-LGd pair. Despite having a low probability of forming loop-like structures, it has the highest probability of extending pairs to loop-like, even though only one loop-like structure has been identified. This is because only two BLA neurons form assemblies with only three LGd neurons. Consequently, only nine possible loop-like structures can be formed, and the associated probability is 0.11, a very high value. This high probability has little meaning because it

would risk assigning too much emphasis to a scarce coordination through loop-like motifs between these areas. For this reason, although we have provided these probabilities in the Supplementary Materials (Sheet “Ranking Comparison”, Column “G”), we have limited in this section the comparison to only the rankings obtained using the probabilities defined in the Methods section, $P_{pairs}^{ext}(Area_i, Area_j)$ and $P_{loop-like}(Area_i, Area_j)$.”

2.

Regarding the results presented in "Hub neurons and loop-like triplets" in the Results section, the authors state that a majority of external hub neurons are from the loop-like population, but this statement does not make much sense without knowing how likely an arbitrary hub neuron belongs to the loop-like population.

Each hub neuron should belong to multiple triplet assemblies, and if at least one of those assemblies is loop-like, this neuron is grouped into the loop-like population.

Conversely, for a neuron to be grouped into the non-loop-like population, all those assemblies must be non-loop-like, which would be a much stricter condition than for being grouped into the loop-like population.

Thus, observing more neurons in the loop-like population than in the non-loop-like population would be a direct consequence of the definition of the populations.

To conclude the positive correlation between the hub-ness and loop-like-ness of a neuron, the authors need to show that there are more hub neurons from the loop-like population than what is expected from the definition of the populations. (I think that the triplet-probability based on identified pair assemblies that I explained above can be used to derive that expectation.)

Furthermore, I think an approach from the other direction would also be necessary: among the neurons in the loop-like population, how many are hub neurons? How is that for the non-loop-like population? Is there a clear difference between the populations in this respect?

Response

We thank the reviewer for raising this issue. In our case, the difference in the number of neurons between loop-like and non-loop-like populations is less than 2%. Specifically, when all areas are combined, we have a total of 7654 neurons in the loop-like population and a total of 7529 neurons in the non-loop-like population. If instead we consider only the 38 areas considered for this analysis, that are the areas with at least 35 neurons in each of the two populations in order to maintain statistical robustness, we have a total of 7022 neurons in the loop-like population and a total of 6472 neurons in the non-loop-like population, that is a 8.5% more in the loop-like population. This is due to the fact that although the condition “belonging to at least one loop-like triplet” may seem easy to satisfy, in reality, the loop-like triplet is a very specific type. Indeed, in addition to having an alternating sequence of neurons belonging to two different areas, it must have a different lag value for each neuron. Individually, the areas show a heterogeneous behavior, with some areas having a greater

number of loop-like neurons and others having a greater number of non-loop-like neurons. For completeness, we have now included a column in the Supplementary Materials showing the number of neurons in each of the two subpopulations for each of the areas analyzed. Regarding the request to approach the issue from the other direction, we thank the reviewer for pointing out that aspect. The histograms in Figure 6 show the distribution of loop-like and non-loop-like neurons in the hub population for each area. This was done for graphical purposes, since in this way, the sum of the two proportions equals the total number of hub neurons, summing to 1, and we were able to use stacked bars. Instead, as explained in the Methods section, the statistical test was used to compare the proportions of hub neurons within the loop-like and non-loop-like populations, as you suggested. To clarify, the contingency matrix used has two groups (loop-like and non-loop-like neurons) and two classes (hub and non-hub neurons). Thus, the test already takes into account the different numerosity of the two groups and evaluates exactly what was requested. However, to confirm our results and, as suggested by the reviewer, we used directly the proportion of loop-like neurons in the analyzed population as the expected probability of having an embedded hub (i.e., a hub neuron belonging to the loop-like population) and we repeated the analysis using another statistical test. Specifically, we have also performed a binomial test using for each area as expected probability for a hub neuron to be considered in the loop-like population, exactly the number of loop-like neurons divided by the sum of the number of loop-like and non-loop-like ones, that is the proportion of loop-like neurons in the analyzed population. As expected, again all areas considered show a statistically significant predominance of hub neurons in the loop-like population compared to the non-loop-like population, as shown in the tables below for both the percentage method (Table 1) and the percentile method (Table 2).

In this way, we have demonstrated that even in this case, there is a clear difference between the populations of loop-like and non-loop-like neurons with respect to the number of hub neurons belonging to one population or the other. However, we prefer not to include this analysis in the main text to avoid overburdening the article while maintaining the statistical analyses already present, which, as explained in the Methods section, are more appropriate in cases like this, where small numbers are present.

We have now added the numbers of hubs (calculated using both the percentage and percentile methods), loop-like, and non-loop-like neurons in the Supplementary Materials. Moreover, we have added the following sentences at the end of the relevant section in the Results:

“It should be noted that the result does not depend on a greater number of loop-like neurons compared to non-loop-like neurons. In fact, when we added up the number of neurons in the loop-like population and compared it with the sum of the non-loop-like neurons in the 38 analyzed areas, we found that the loop-like population was only 8.5% larger. Furthermore, the different sizes of the two subpopulations were considered in the statistical analysis. For enhanced readability, the absolute numbers of hubs (calculated using the percentage and percentile methods), as well as the numbers of loop-like and non-loop-like neurons, are reported in the Supplementary Materials (sheet “*Hub Statistics*”, columns “*K–P*”).”

Areas	p-value	n° hubs in Loop-like	n° neurons Loop-like	n° hubs in Non-loop-like	n° neurons Non-loop-like
ACA	6,16344E-19	169	228	12	147
ACB	3,2204E-07	23	67	4	115
APN	5,32798E-07	65	99	11	70
CA1	1,60964E-48	206	355	17	400
CA3	2,38579E-18	67	113	6	152
CP	6,06773E-12	73	160	20	209
DG	1,58773E-34	138	249	10	280
ILA	4,48212E-11	50	119	2	96
LD	1,11539E-06	55	83	26	121
LGd	6,65648E-18	54	108	14	268
LP	4,16334E-14	55	96	31	285
LS	1,15116E-13	121	219	0	63
LSr	2,85259E-17	103	214	3	155
MB	1,41372E-17	180	335	13	193
MD	0,000418486	31	139	0	42
MG	1,17487E-08	32	77	9	148
MOp	1,14352E-14	219	442	5	124
MOs	4,65743E-20	154	277	25	247
MRN	1,52323E-14	258	538	10	146
ORB	4,69402E-06	14	107	3	277
PL	6,88965E-14	130	270	9	134
PO	0,002373028	31	101	19	148
POL	1,14963E-09	43	84	1	62
POST	1,66356E-10	90	135	7	73
RSP	1,65255E-10	52	94	18	163
SCig	2,04993E-06	56	303	1	101
SCm	1,62148E-05	65	146	8	74
SNr	1,05509E-07	25	76	0	69
SSp	3,94863E-11	142	258	12	109
SUB	6,36853E-28	215	336	8	221
TH	5,0625E-32	164	283	32	367
TT	7,7708E-05	27	59	2	42
VISa	1,23636E-11	38	60	11	138
VISam	4,1883E-27	178	288	27	275
VISl	1,11642E-22	90	126	11	194
VISp	2,03512E-17	87	166	41	474
VISpm	2,43323E-16	69	105	19	192
VPL	9,06968E-19	67	107	0	98

Table 1. The percentage method is used to detect hub neurons.

Areas	p-value	n° hubs in Loop-like	n° neurons Loop-like	n° hubs in Non-loop-like	n° neurons Non-loop-like
ACA	2,43893E-15	72	228	0	147
ACB	0,002042385	8	67	1	115
APN	0,000390184	16	99	0	70
CA1	1,19671E-23	85	355	5	400
CA3	1,84138E-11	29	113	0	152
CP	0,004273089	27	160	14	209
DG	2,44249E-15	49	249	1	280
ILA	0,000889417	12	119	0	96
LD	4,20309E-07	26	83	4	121
LGd	1,46123E-11	20	108	0	268
LP	7,47083E-05	14	96	7	285
LS	0,003199224	26	219	0	63
LSr	2,617E-07	29	214	0	155
MB	1,24221E-05	26	335	0	193
MD	0,032119233	16	139	0	42
MG	7,10349E-09	22	77	2	148
MOp	9,24529E-10	119	442	3	124
MOs	2,85082E-06	28	277	2	247
MRN	1,22644E-05	49	538	0	146
ORB	1,90077E-06	16	107	4	277
PL	0,000278895	21	270	0	134
PO	0,005124059	10	101	2	148
POL	0,012833081	9	84	0	62
POST	2,64835E-13	68	135	0	73
RSP	0,000649711	15	94	5	163
SCig	7,66281E-05	43	303	1	101
SCm	0,011431946	12	146	0	74
SNr	0,001103495	11	76	0	69
SSp	1,9197E-10	117	258	8	109
SUB	5,65512E-15	85	336	0	221
TH	4,10783E-15	47	283	2	367
TT	0,046506823	7	59	0	42
VISa	0,000156701	9	60	1	138
VISam	7,80617E-14	50	288	1	275
VISl	2,44249E-15	45	126	3	194
VISp	7,77156E-16	30	166	2	474
VISpm	1,95891E-17	37	105	0	192
VPL	1,87447E-07	24	107	0	98

Table 2. The percentile method is used to detect hub neurons.

Minor comments:

1.

The authors often refer to their previous study about ZI, but there is no explanation about what ZI is.

Please introduce this term properly.

Response

We have now defined ZI as zona incerta in the manuscript where we first introduced the term and added it to Table S1, "Clarification of Area Acronyms," in the Supplementary Information file.

2.

The authors compute many types of "probability", which is defined as the fraction of a specific type of pairs/triplets identified by the CAD algorithm among all possible pairs/triplets.

Technically, this should be termed as "density" rather than "probability".

For reporting how many pairs/triplets are identified among all possible cases, using the term "density" simply suffices, while using the term "probability" implies a certain assumption of a stochastic process about the formation (by the brain) or the measurement (by the experimenter) of the pairs/triplets, which I do not think necessary in the context of the present study.

I would suggest using "density" instead of "probability" throughout the manuscript.

Otherwise, the authors should clarify what stochastic process the "probability" is meant for.

Response

We thank the reviewer for this thoughtful comment on terminology. In the manuscript, we do not intend to imply that any generative stochastic process governs the formation of assemblies in the brain when we use the term "probability." Instead, we use "probability" in the precise sense of an empirical probability/proportion defined on a finite, discrete sample space. Specifically, given a set of n neurons and fixed CADopti parameters (i.e., the set of possible time bins and lags), we consider the set of all possible pairs or triplets as the sample space. If one draws a pair or triplet at random from this set, the reported quantity is the probability that the selected combination is classified as an assembly by CADopti. In other words, this is the fraction of CAD-detected pairs or triplets among all possible pairs or triplets. This definition does not require specifying an underlying stochastic process and is fully determined by the finite combinatorial space and the detection rule applied to the data.

We agree that the term "density" is sometimes used informally to indicate a normalized count. However, in a strict statistical sense, "density" more commonly refers to probability density functions for continuous variables, which is not the case here as we are counting discrete combinations. For clarity and consistency, we prefer to use the term "probability" while explicitly stating the above interpretation. Accordingly, we have added a clarification in the Methods to make explicit that these "probabilities" should be understood as empirical probabilities (i.e., proportions) over all possible neuron pairs/triplets, conditioned on the

chosen algorithm and parameter set. We also retain this terminology to remain consistent with our previous work, in which the same normalized quantities were defined and used to enable comparability across recordings and regions with different numbers of neurons.

To clarify we have included the following sentence in the Method section:

“Throughout the manuscript, "probability" refers to an empirical probability, or a proportion, defined over the finite set of all possible neuron pairs or triplets. More specifically, given a recording and fixed CADopti parameters (i.e., possible time bins and lags), probability denotes the likelihood that a randomly selected pair or triplet from all possible combinations will be classified as an assembly by CADopti. This probability is calculated by dividing the number of CAD-detected pairs or triplets by the total number of possible pairs or triplets.”

3.

About the color map used in Fig. 3A-B and Fig. 4A-B for probability, it would be more intuitive and graphically appealing to me if the brighter colors corresponded to higher probabilities and vice versa.

Response

We have modified figures 3 and 4 A-B (lower triangles) by changing the color map as suggested.

4.

Also about these figures, the relation between (Area_i, Area_j) and (row, column) of the matrix needs to be explained in the caption.

Response

We have now added to the captions of Figures 3 and 4 the relationship between the row/column elements (i,j) and the corresponding type of assembly between Area_i and Area_j, for which the heatmap shows the associated probability, thanks.

5.

In page 16, line 1, "best and worst" should be rephrased to "top and bottom".

Response

We have rephrased as suggested replacing best and worst with top and bottom.

6.

In the last two paragraphs of page 16, please describe the findings with the actual values of the relevant measures, citing them from the supplementary spreadsheet.

Response

We have now added the actual values of the relevant measures, taking them from the supplementary spreadsheet.

7.

In the second paragraph of page 18, "the last 30 positions" should be rephrased to "the bottom 30 positions".

Response

We have rephrased as suggested, replacing "worst" with "bottom".

8.

In the second paragraph of page 20, "the external regions they were recorded with" should be rephrased to "the other simultaneously recorded regions".

Response

We have rephrased as suggested, replacing "the external regions they were recorded with" with "the other simultaneously recorded regions".

9.

In the caption to Fig. 6, the explanation about "on the right (or left) side of each histogram" is not correct: that is inverted for the external hub histogram.

Also, "external hubs (left histogram)" is not correct: it is the right histogram, if I understand the figure correctly.

Response

We have corrected external hubs (left histogram) with external hubs (right histogram).

10.

In page 23, line 1, what "dyadic coordinative relationships" means is not clear.

Response

We have rephrased using the terminology used in the manuscript: "higher-than-average number of inter-regional pairs with neurons..."

11.

In the second paragraph of page 23, the term "embedded hub" should be explicitly defined before it is first used. (It is not clear enough from the description in the previous paragraph.)binary regression

Response

We had improved the definition provided on page 23:

“One main aim of the present study was to test whether what we defined as “embedded hubness,” that is, the association between hub neurons and loop-like motifs of inter-area interaction, where embedded neuron hubs are part of loop-like motifs, was specific to the ZI or was a common modality of information processing”

END OF COMMENTS

Dear Professor Genovesio,

Re: JP-RP-2026-289827R1 "**Neurons embedded in loop-like motifs act as central hubs for brain-wide integration**" by Fabrizio Londei, Giulia Arena, Lorenzo Ferrucci, Francesco Siano, Encarni Marcos, Francesco Ceccarelli, and Aldo Genovesio

We are pleased to tell you that your paper has been accepted for publication in The Journal of Physiology.

Yours sincerely,

Richard Carson
Senior Editor
The Journal of Physiology

IMPORTANT POINTS TO NOTE FOLLOWING ACCEPTANCE OF YOUR PAPER:

- **IMPORTANT NOTICE ABOUT OPEN ACCESS:** To assist authors whose funding agencies mandate immediate public access to published research findings, The Journal of Physiology allows authors to pay an Open Access (OA) fee to have their papers made freely available immediately on publication.

- You can help your research get the attention it deserves! Check out Wiley's free Promotion Guide for best-practice recommendations for promoting your work at: www.wileyauthors.com/eoo/guide. You can learn more about Wiley Editing Services which offers professional video, design, and writing services to create shareable video abstracts, infographics, conference posters, lay summaries, and research news stories for your research at: www.wileyauthors.com/eoo/promotion.

- If you would like to receive our 'Research Roundup', a monthly newsletter highlighting the cutting-edge research published in The Physiological Society's family of journals (The Journal of Physiology, Experimental Physiology, Physiological Reports, The Journal of Nutritional Physiology and The Journal of Precision Medicine: Health and Disease), please click this link, fill in your name and email address and select 'Research Roundup': <https://www.physoc.org/journals-and-media/membernews>

EDITOR COMMENTS

Reviewing Editor:

Comments to the Author:

Thank you very much for revising your submission to The Journal of Physiology. All concerns have been addressed. I congratulate the authors on their nice work.

REFEREE COMMENTS

Referee #1:

The authors have addressed all of my comments. I have no further comments.

Referee #2:

The authors thoroughly discussed the issues raised in my previous report.

My concerns there were all properly addressed, and the manuscript has been revised accordingly.

I have no remaining concerns about the manuscript.